# ATP synthase hexamer assemblies shape cristae of *Toxoplasma* mitochondria

Alexander Mühleip [1,2,4], Rasmus Kock Flygaard [1,2,4], Jana Ovciarikova[3], Alice Lacombe[3], Paula Fernandes [1,2,3], Lilach Sheiner [3✉] & Alexey Amunts [1,2✉]

Mitochondrial ATP synthase plays a key role in inducing membrane curvature to establish cristae. In Apicomplexa causing diseases such as malaria and toxoplasmosis, an unusual cristae morphology has been observed, but its structural basis is unknown. Here, we report that the apicomplexan ATP synthase assembles into cyclic hexamers, essential to shape their distinct cristae. Cryo-EM was used to determine the structure of the hexamer, which is held together by interactions between parasite-specific subunits in the lumenal region. Overall, we identified 17 apicomplexan-specific subunits, and a minimal and nuclear-encoded subunit-*a*. The hexamer consists of three dimers with an extensive dimer interface that includes bound cardiolipins and the inhibitor IF$_1$. Cryo-ET and subtomogram averaging revealed that hexamers arrange into ~20-megadalton pentagonal pyramids in the curved apical membrane regions. Knockout of the linker protein ATPTG11 resulted in the loss of pentagonal pyramids with concomitant aberrantly shaped cristae. Together, this demonstrates that the unique macromolecular arrangement is critical for the maintenance of cristae morphology in Apicomplexa.

---

[1] Science for Life Laboratory, Department of Biochemistry and Biophysics, Stockholm University, 17165 Solna, Sweden. [2] Department of Medical Biochemistry and Biophysics, Karolinska Institute, 17177 Stockholm, Sweden. [3] Wellcome Centre for Integrative Parasitology, University of Glasgow, Glasgow, UK. [4]These authors contributed equally: Alexander Mühleip, Rasmus Kock Flygaard. ✉email: lilach.sheiner@glasgow.ac.uk; amunts@scilifelab.se

F-type ATP synthases are energy-converting membrane protein complexes that synthesize adenosine triphosphate (ATP) from ADP and inorganic phosphate. These universal enzymes function by using the energy stored in an electro-chemical potential across the bioenergetic membrane by rotary catalysis[1,2]. The soluble $F_1$ subcomplex and membrane-bound $F_o$ subcomplex together form the $F_1F_o$ ATP synthase monomer, which is found in bacteria and chloroplasts[3,4]. In mitochondria, $F_1F_o$ ATP synthase resides in the crista membrane where it is known to form dimers, which can further assemble into rows critical for inducing the membrane curvature and maintaining membrane potential and morphology[5–10].

The main driving force for the synthesis of ATP in mito-chondria is the membrane potential[11], which has been shown to be higher in the cristae lumen than in the adjacent inter-membrane space[12]. Cristae shaping has been shown to depend on the assembly of ATP synthase dimers into dimer rows, which is the basis for energy conversion in all mitochondria studied to date[13]. However, the molecular interactions that convey the membrane-shaping properties of the oligomeric ATP synthase are poorly understood. Furthermore, structural data has shown that cristae morphology varies between eukaryotic lineages[13].

The infectious apicomplexan parasite *Toxoplasma gondii*[14] is commonly used as a model organism for the malaria-causing agent *Plasmodium* spp[15]. These parasites have a unique bulbous cristae morphology, which differs substantially from the lamellar cristae of their mammalian hosts[16,17]. The underlying mechanism for the bulbous cristae is unknown. Loss of ATP synthase is accompanied by parasite death and defects in cristae abundance in the *T. gondii* stage responsible for acute toxoplasmosis[18], and results in the death of the *Plasmodium* mosquito form responsible for malaria spread[19]. Here, we investigate the mechanism for the generation of the unique cristae in the Apicomplexa, using a combination of single-particle cryo-EM, cryo-ET and sub-tomogram averaging. We first report cristae-embedded ATP synthase hexamers arranged in pentagonal pyramids in the wild type, then identify a key subunit for the assembly, and finally characterise mutant cells with a generated knockout of this subunit.

## Results

### Structure of the hexameric ATP synthase and its herein identified elements

A large-scale preparation of *T. gondii* tachyzoite mitochondria and subsequent mild solubilisation with digitonin resulted in the isolation of intact ATP synthase complexes, which we identified as native hexamers. We then performed solubili-sation with *n*-dodecyl-β-D-maltoside (β–DDM) that resulted in dissociation of the hexamers into dimers. Both oligomeric forms were subjected to cryo-EM structure determination (Fig. 1, Sup-plementary Figs. 1 and 2). Masked refinements of the ATP syn-thase dimer resulted in maps of the membrane region, the OSCP/$F_1$/c-ring complex, the rotor and the peripheral stalk, ranging in resolution from 2.8 to 3.5 Å (Supplementary Figs. 1 and 3), thus allowing de novo modelling of the respective regions. Refinement into a 2.9-Å resolution consensus map allowed model construc-tion of the entire ATP synthase dimer (Fig. 1a, b and Supple-mentary Table 1). The 1.85-MDa complex consists of 32 different subunits, of which only 15 are canonical with structural equiva-lents in other phyla. Homolog searches of 17 noncanonical sub-units revealed them to be largely conserved in mitochondriate Apicomplexa including *Plasmodium* parasites, and in the related phyla of chromerids and perkinsozoa, suggesting that the herein described architecture is likely representative of myzozoans (Supplementary Fig. 4). Thus, following a species-specific nomenclature established in protozoan ATP synthases[20–22], we

term the 17 apicomplexan-conserved *T. gondii* subunits ATPTG1-17 (TG for *T. gondii*), with ATPTG1, ATPTG7, and ATPTG16 identified directly from the cryo-EM map (Supple-mentary Fig. 5a and Supplementary Table 2).

The apicomplexan-conserved subunits and extensions of the canonical subunits constitute a membrane-embedded $F_o$ sub-complex, which ties the two $F_1$/c-ring subcomplexes together at the angle of 19° (Fig. 1a, b). This is in stark contrast to ~100° found in the yeast and mammalian ATP synthase dimers[23,24], suggesting the narrow-angle parasite dimer induces substantially less membrane curvature. The enlarged *T. gondii* $F_o$ subcomplex differs markedly in its overall architecture from other ATP synthase structures. It displays distinct structural features including a peripheral matrix-exposed part that we term 'wing' region and a 360-kDa lumenal region (Fig. 1b). The $F_o$ periphery contains several compact folds, including three coiled-coil-helix-coiled-coil-helix domain (CHCHD) containing proteins ATPTG7-9; a thioredoxin-like fold in ATPTG4; and ubiquitin-like fold in subunit-*k* (Supplementary Fig. 5b).

The apicomplexan-conserved $F_o$-subunit ATPTG11 extends from the lumenal region and plugs the central cavity of the c-ring. This is mediated by the short N-terminal amphipathic helix of ATPTG11 (Ala9-Leu17), which is sequence-conserved in Api-complexa (Fig. 2a, b). The interface is located on the border of the detergent belt and dominated by hydrophobic residues of ATPTG11 pointing towards the inside of the c-ring, which is unlikely to inhibit the rotation of the rotor (Fig. 2a, c). Protein density on the inside of the c-ring, as suggested for the porcine complex[24], was not observed.

Masked refinement of the hexamer membrane region resulted in a 4.8-Å resolution map (Supplementary Fig. 2). Refining three copies of the dimer model into the hexamer map resulted in a good fit and showed that it forms a cyclic trimer of dimers. No additional subunits or substantial conformational changes of the dimer units were found in the hexameric assembly (Fig. 1c, d and Supplementary Movie 1). The $C_2$ symmetry axis through the dimer is tilted 22° with respect to the $C_3$ axis in the hexamer, thereby bringing into proximity the lumenal regions, which extend 80 Å from the membrane.

### Parasite-specific subunits form a dimer interface that includes IF$_1$ and bound cardiolipins

The structure of the *T. gondii* ATP synthase reveals that the unusual architecture of the dimer is generated by the peripheral stalks that are laterally offset, extending away from the central dimer axis (Fig. 1a, b). This architecture does not allow the formation of the conventional dimerization interface of type-I ATP synthases found in animals and yeast (Supplementary Fig. 5i, j), in which peripheral stalks extend along the dimer long axis[13]. We therefore examined the dimerization interface, which is formed by the apicomplexan subunits and extensions of the canonical $F_o$ subunits. Those elements involve eleven proteins from each monomer that con-tribute more than 7000 Å$^2$ of buried surface area, making the interface substantially larger than in mammalian, yeast and algal ATP synthase structures (Supplementary Fig. 5c–j).

The dimerization interface in the membrane and lumenal regions is governed by homotypic interactions between symmetry-related subunits, most of which extend deep into both monomers. Subunit-*b* contains two transmembrane helices, each binds one of the symmetry-related subunit-*a* copies, which are therefore linked by four transmembrane helices (Fig. 2a, b). In addition, two cardiolipins are found on the matrix side, forming specific protein-lipid interactions bridging the two copies of subunit-*b* and subunit-*f* (Fig. 3a). This cardiolipin pair is sequestered in the $F_o$ subcomplex with no apparent path to the

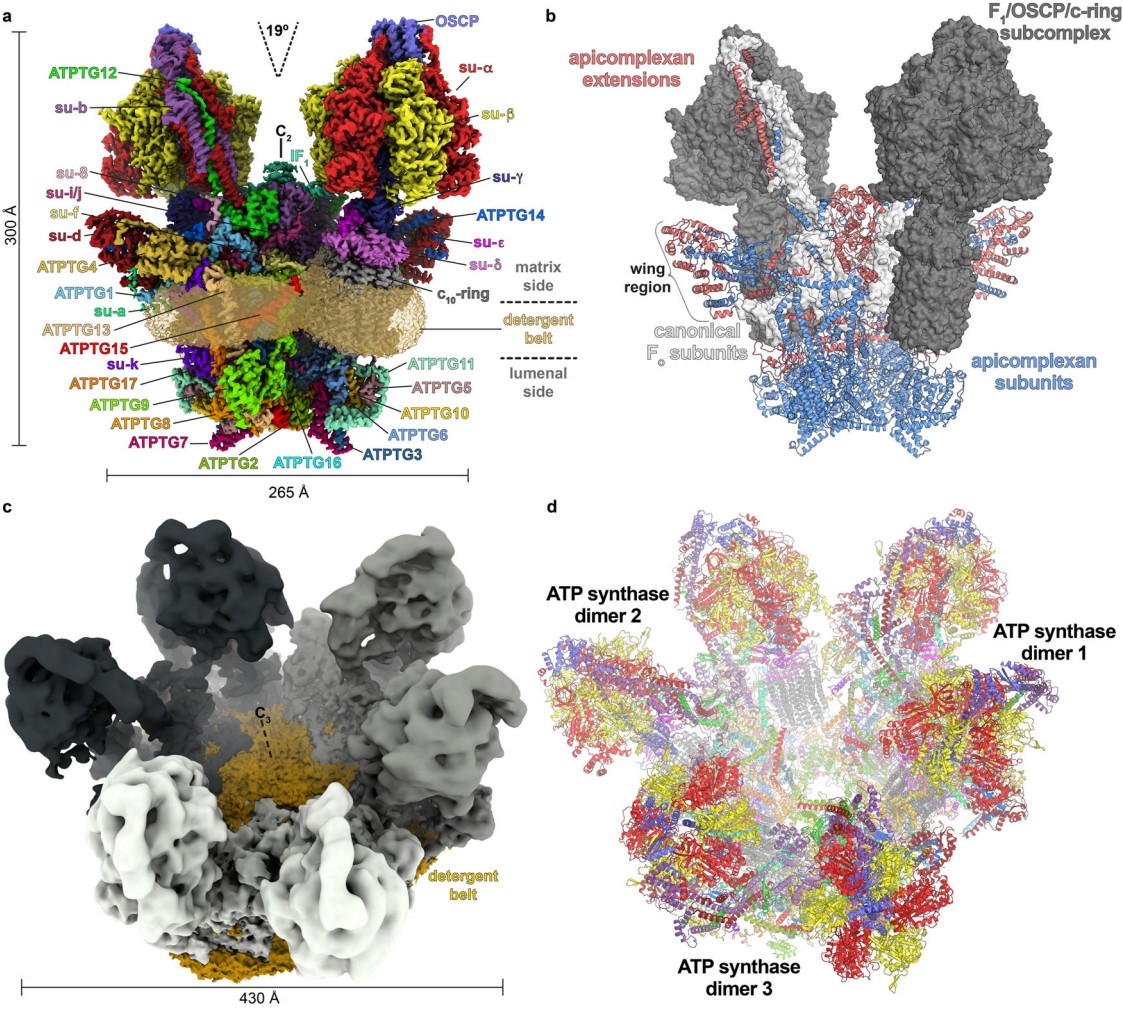

**Fig. 1 Overall architecture of *T. gondii* ATP synthase dimer and hexamer. a** The composite cryo-EM map of the dimer highlights a small dimer angle and large lumenal region. **b** The atomic model of the dimer with highlighted apicomplexan-specific structural components responsible for the specific mode of dimerization. **c** Cryo-EM map of the hexamer showing an assembly as trimer of dimers. **d** Atomic model of the hexamer with individually coloured subunits.

bulk membrane, suggesting a structural role. An additional 15 cardiolipins and 12 other phospholipids were found to mediate a network of interactions throughout the membrane region (Supplementary Fig. 6a). These native lipids are primarily bound in two vestibules within the $F_o$ subcomplex (Fig. 3a) with the lipid head groups mediating charged interactions between numerous subunits (Supplementary Fig. 6b–e), which indicates a contribution to the stability of the complex.

The inhibitor protein of ATPase activity $IF_1$ is bound to subunit-*b*, contributing to the $F_o$ dimer interface with its C-terminal helix extending from $F_1$ to interact with subunit-*b*' of the neighbouring monomer (Fig. 3b). $IF_1$ in our structure is bound exclusively to the α/β-interface facing the dimer interface (Fig. 3b and Supplementary Fig. 7a–d, f), thereby locking it in the ADP-bound state ($β_{DP}$). The N-terminal $IF_1$ region that contacts the central stalk in the bovine complex[25] is absent in our structure (Supplementary Fig. 7b). Because central stalk rotation and conformational changes in the catalytic sites of $F_1$ are interdependent, the sterically restrictive $IF_1$ binding in *T. gondii* to only one of the three catalytic sites results in the trapping of the ATP synthase in a single rotational state in both the dimer and hexamer. In our cryo-EM maps, $IF_1$ is contiguous with $F_o$-associated density extending to the $C_2$-symmetry axis, thus linking the two $F_1$ monomers (Fig. 2d, Supplementary Fig. 7f). We assign it to the unmodeled C-terminal region of $IF_1$, which

has previously been characterised as a homo-oligomerisation domain in mammals[26]. This bridging of two $F_1$ monomers is intra-dimeric, which is different from the mammalian ATP synthase tetramer, where bridging occurs between the two neighbouring dimers[24,27] (Supplementary Fig. 7e, f).

**Evolutionary and functional aspects of a minimal and nuclear-encoded subunit-*a*.** We assigned subunit-*a* by locating topologically conserved transmembrane helices of the canonical subunits *b*, *d*, *f*, *i/j*, *k*, and 8 (Fig. 2a, b and Supplementary Fig. 8d). Based on the sequence identified directly from the cryo-EM map, we found that *T. gondii* subunit-*a* is encoded in the nucleus, and not in mitochondria as in most other organisms. Thus, in *T. gondii*, all ATP synthase subunits are nuclear-encoded (Supplementary Table 2). In addition, unlike in the canonical six-helix ($H1-6_a$) fold, which is conserved in bacteria, chloroplasts and other mitochondria[4,28–30], the subunit-*a* in *T. gondii* lacks $H1-4_a$, and only the horizontal $H5_a$ and $H6_a$ are found. They interact with the c-ring at the rotor-stator interface (Fig. 2a, b; Supplementary Fig. 8a, b). This is the smallest subunit-*a* structure reported to date.

The unmodelled sequence that would make up the canonical transmembrane $H1-4_a$, corresponds to a mitochondrial targeting sequence with a predicted cleavage site located N-terminally of

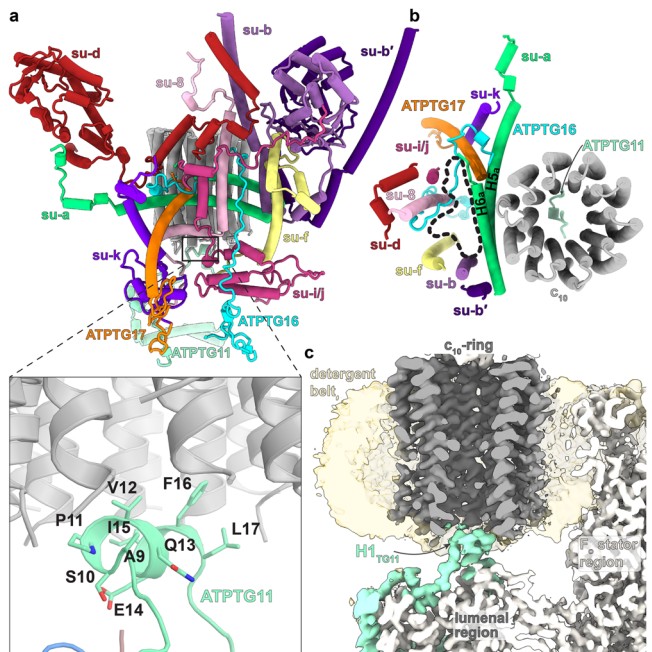

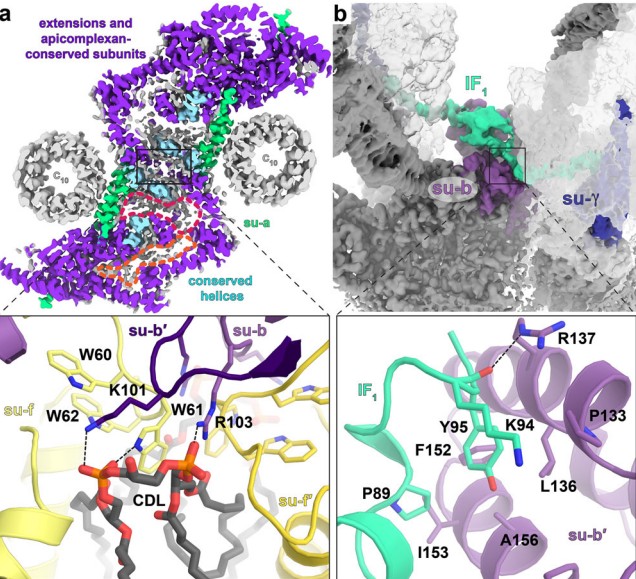

**Fig. 2 Conserved and apicomplexan-specific $F_o$ components of hexameric ATP synthase. a** The canonical $F_o$ subunits *b, d, f, i/j, k,* and *8* contain apicomplexan-specific extensions contributing to a large $F_o$. Subunit-*a* contains only the conserved H5-6$_a$, and ATPTG16, ATPTG17 partially replace the missing H1-4$_a$. Inset shows N-terminal helix of ATPTG11 forming a parasite-specific rotor-stator interface with the lumenal side of the c-ring. **b** Top view of rotor-stator interface. The absence of H1-4$_a$ separates subunit-*a* from several canonical $F_o$-subunits. Resulting lipid-filled $F_o$ void outlined (black dash). **c** Cross section through the $F_o$ region of the map. ATPTG11 extends from the lumenal region to plug the c-ring through interactions with H1$_{TG11}$.

**Fig. 3 Parasite-specific subunits and resolved lipids at the dimer interface. a** $F_o$ map cross-section showing apicomplexa-specific (purple) and conserved (light blue) subunits, as well as lipid vestibules (red, orange). The apicomplexa-specific subunits scaffold the $F_o$ architecture. Close-up inset shows protein-cardiolipin (CDL) contact at dimer interface with subunits *b* and *f* interacting via a tightly bound cardiolipin. **b** IF$_1$ dimer (green density) binds to $F_o$ (dark grey density) and $F_1$ of both monomers (transparent grey), linking them together. Close-up inset shows IF$_1$ interactions with subunit-*b* (violet).

H5$_a$, thereby causing the truncation (Supplementary Fig. 8c). Thus, compared to its mitochondria-encoded homologs, *T. gondii* subunit-*a* displays a reduced overall hydrophobicity, which we found to be conserved in Apicomplexa (Fig. 4a). A similar observation for different mitochondrial membrane proteins has been proposed to enable mitochondrial protein targeting following gene transfer to the nucleus[31–33].

The missing interactions of truncated H1-4$_a$ are compensated by lipids and apicomplexan subunits and extensions surrounding the canonical subunits, anchoring them to the enlarged $F_o$ region and the wing region (Figs. 2b and 3a). Thus, the minimal *T. gondii* subunit-*a* exemplifies an evolutionary mechanism that combines subunit truncation and reduced hydrophobicity with structural compensation that allowed gene transfer. Together, our analyses illustrate how the substantial mitochondrial genome reduction occurred in apicomplexan parasites, retaining only three mitochondrial genes, while maintaining functional mitochondrial energy conversion.

**The IF$_1$-locked rotational state reveals salt bridge formation at the rotor-stator interface.** In addition to minimal architecture and evolutionary insight, the subunit-*a* structure also reveals its interactions with the c-ring. The IF$_1$-arrested structure, in which ATP synthases are locked in a single rotational state, allowed us to obtain a map of the rotor-stator interface at 3.5 Å (Supplementary Fig. 1d), resolving both the c-ring and H5-6 of subunit-*a*, where proton transfer occurs. Mechanistically, the essential arginine on H5$_a$ (Arg166 in *T. gondii*) is thought to be responsible for deprotonation of the conserved glutamate on the c-ring

(Glu150 in *T. gondii*)[29]. Translocating protons enter $F_o$ via a lumenal access channel, are transferred to the protonatable glutamate on the c-ring and released via a matrix channel[34,35]. While our cryo-EM map does not display unambiguous density for Glu150, previous X-ray crystal structures have shown that this side chain can adopt an open unprotonated or closed proton-locked rotamer[36,37]. Both formation and absence of a salt bridge between the arginine and glutamate have been observed in different structures[29,38,39], including a suggestion of a bridging water molecule[40].

Our structure indicates that in the open conformation Glu150 is within 2.3 Å distance from the juxtaposed Arg166, allowing the formation of a salt bridge (Fig. 4b). The rotor-stator interface surrounding the Arg166/Glu150 pair is more hydrophobic compared to other structures, with subunits *a* and *c* contributing a total of eight aromatic residue side chains (Fig. 4b, Supplementary Fig. 8e). Thus, the tight hydrophobic interface between the decameric c-ring and subunit-*a* in *T. gondii* is consistent with a direct, rather than water-mediated Arg/Glu interaction.

We traced two cavities in the $F_o$ subcomplex corresponding to the proton half-channels on the lumenal and matrix sides (Fig. 4c, d, Supplementary Fig. 8g). The lumenal proton half-channel displays a hydrophilic entrance between subunit-*a* and ATPTG2 facing towards the c-ring (Fig. 4d, Supplementary Fig. 8f). Inside the membrane, the lumenal channel is lined by membrane-inserted loops of ATPTG2 and ATPTG3 and the C-terminal transmembrane helix of subunit-*b* (Fig. 4d). The channel extends through the only acidic patch between H5$_a$ and H6$_a$ near a conserved glutamate (Glu201), which is thought to mediate proton transfer to the c-ring (Supplementary Fig. 8f)[29]. The matrix half-channel locates to a hydrophilic region between subunits *a, d,* ATPTG16, ATPTG17 and extends into the membrane region towards R159 of H5$_a$, which is widely conserved (Fig. 4c). Remarkably, the C-terminus of ATPTG16 contributes

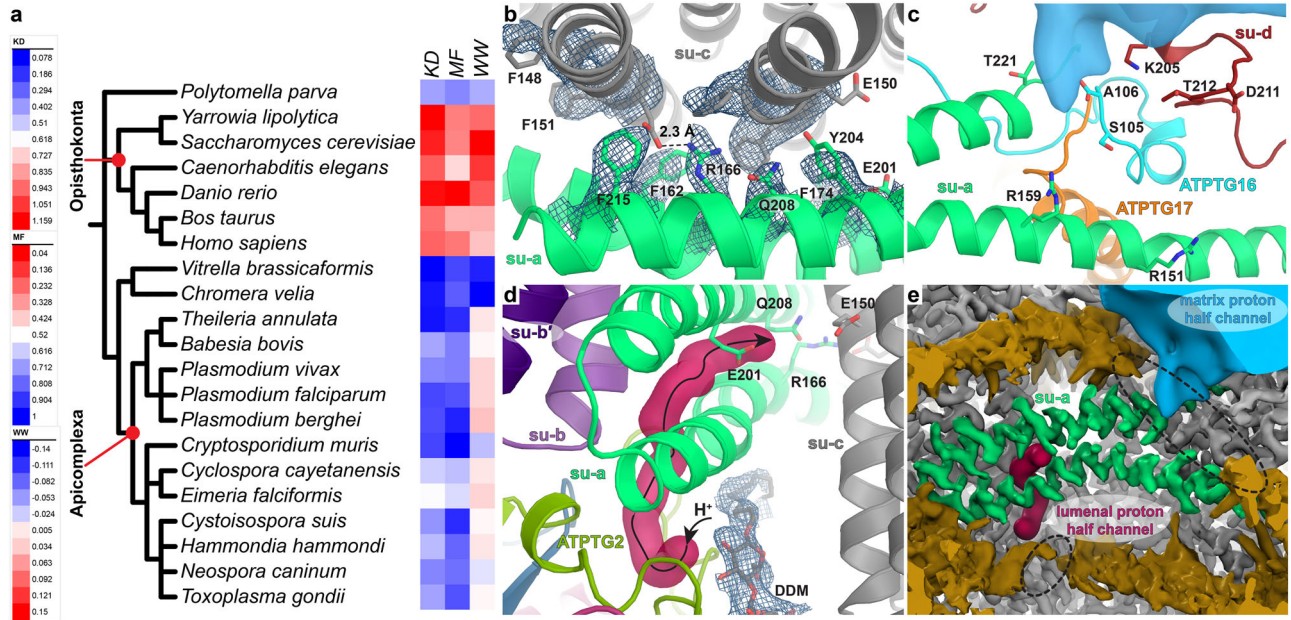

**Fig. 4 The minimal subunit-*a* is parasite-conserved and forms a salt bridge at the rotor-stator interface. a** Heat map indicating the average hydrophobicity of subunit-*a* in divergent organisms, calculated as the grand average of hydropathy (KD)[84] or according to the Moon-Fleming (MF)[85] or Wimley-White (WW)[86] hydrophobicity scales. The nuclear-encoded subunit-*a* of apicomplexan parasites, as well as the related chromerid alveolates *C. velia* and *V. brassicaformis* show a reduced hydrophobicity compared to the mitochondria-encoded subunit-*a* homologs of animals and fungi. Reduced hydrophobicity is also found in subunit-*a* of the green alga *P. parva*, which is also nuclear encoded[87] and lacks TM helix 1. **b** Top view of the subunit-*a/c* interfaces. The central arginine/glutamate pair is within interaction distance and enclosed by six aromatic residues. **c** Close-up view of the matrix half-channel (blue) with hydrophilic residues of subunits *a*, *d* and the C-terminus of ATPTG16 indicated. **d** Lumenal half-channel (burgundy) with proposed proton path to c-ring (black arrows). The channel entrance is occupied by a β-DDM molecule. **e** Proton half-channels shown in red (lumenal) and blue (matrix) colours and compared to gaps (dotted black circles) in detergent density (dark gold) of the cryo-EM density map of the dimer.

the only nearby carboxylate group, likely serving an equivalent role to acidic side chains thought to mediate proton release in ATP synthases of other organisms (Fig. 4c)[24,28,29,38,41].

The lateral offset between the two proton half-channels is also evident in the density map, where discontinuation of the detergent belt matches the positions of the two half-channels in support of an aqueous environment for proton translocation (Fig. 4e, Supplementary Fig. 8g). Taken together, both half-channels are partially lined by apicomplexan-specific subunits resulting in a divergent structure and likely the involvement of different residues in proton translocation compared to structures from other organisms.

**Peripheral stalk subunit-*b* contains a structural motif found in the mammalian subunit F6**. The peripheral stalk extends from the membrane-embedded part of $F_o$ and attaches to the tip of $F_1$, holding it stationary against the torque of the central stalk. In *T. gondii* the peripheral stalk is composed of subunit-*b*, *d*, ATPTG12 and OSCP (Fig. 1a, Supplementary Fig. 9b). The attachment to $F_1$ is mediated through OSCP, which adopts a fold conserved in prokaryotic and eukaryotic homologs (Supplementary Fig. 9a). Subunit-*b* displays structural similarity with its bacterial, algal and mammalian counterparts, engaging in conserved interactions with the C-terminal domain of OSCP as observed in other structures. Compared to the yeast and mammalian ATP synthases, *T. gondii* displays an augmented peripheral stalk structure with extensions in subunit-*b*, subunit-*d* and the additional ATPTG12 (Supplementary Fig. 9b). Unlike yeast and porcine[24,39], neither subunit-*f* nor 8 (A6L in mammals) contribute to the peripheral stalk. Instead, the apicomplexan-conserved subunit ATPTG12 forms extensive interactions with

subunit-*b* and *d* throughout the peripheral stalk structure (Supplementary Fig. 9b).

Interestingly, peripheral stalk subunit F6 (subunit *h* in yeast) is not found in *T. gondii* ATP synthase. Instead, the C-terminal extension of subunit-*b*, adopts a fold that structurally resembles subunit F6/*h* and provides supporting interactions with the long subunit-*b* helix (Supplementary Fig. 9c). Both, the yeast subunit-*h* (on non-fermentable carbon sources) and the augmented *T. gondii* subunit-*b* are essential[18,42], suggesting a critical role in peripheral stalk assembly.

**Formation of the ATP synthase hexamer involves two contact sites in the lumenal region**. Next, we asked how ATP synthase dimers interact in the hexamer structure to form the cyclic trimer of dimers. The hexamer model shows that each of the three dimer-dimer interfaces contributes ~1211 $Å^2$ to hexamer contacts. Those contacts holding the hexamer together are found in two separate sites in the lumenal regions, which form a triangular subcomplex (Fig. 5a).

In the first site, three copies of the subunit ATPTG9 are arranged around the $C_3$-symmetry axis, directly beneath the central lipid bilayer (Supplementary Fig. 2e, f). ATPTG9 contains two CHCHDs with cysteine pairs positioned to form disulfide bonds. CHCHD-containing subunits were reported to play a role in the assembly of Complex IV[43]. In our structure, H2 of the first CHCHD in one ATPTG9 copy interacts with $H5_{TG9}$ and the loop connecting β2 and $H5_{TG9}$ of a neighbouring ATPTG9 (Fig. 5a). Both interacting structural elements are predominantly hydrophilic, consistent with their solvent-accessible location in the lumen.

In the second site, located at the periphery of the lumenal region, ATPTG11 establishes a network of five interacting subunits. A helix hairpin of ATPTG11, containing a central cysteine pair, extends

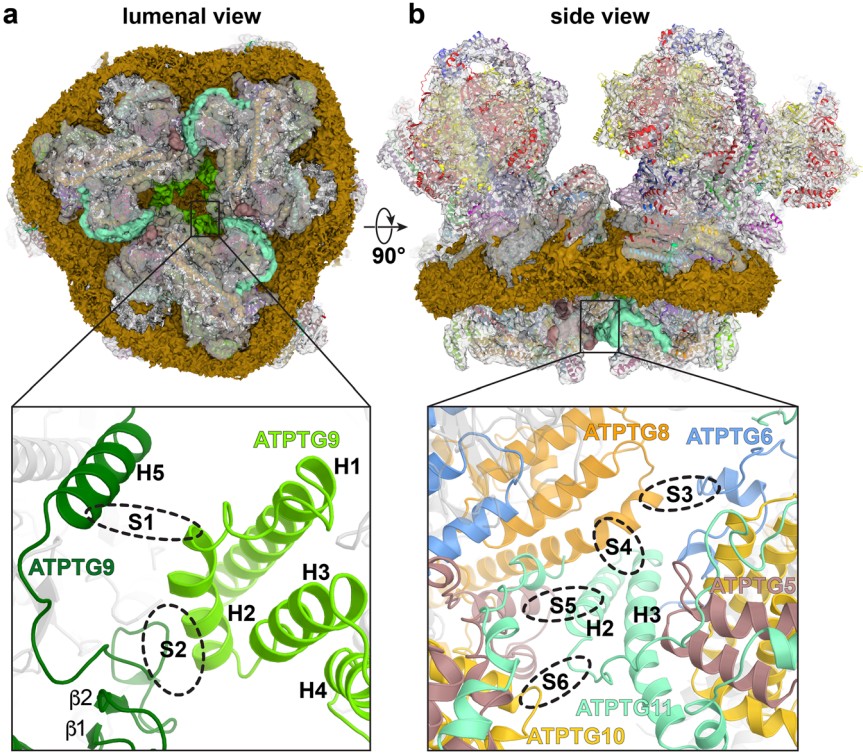

**Fig. 5 Structure of *T. gondii* ATP synthase hexamer reveals two lumenal F$_o$ contact sites between neighbouring dimers. a** First contact site is shown on the ATP synthase hexamer composite map viewed from the lumen with central lipid bilayer and surrounding detergent belt (gold). The lumenal regions of the three dimers interact to form a triangular complex (bottom panel). Three copies of the CHCHD protein ATPTG9 form homotypic interactions. **b** Second contact site is shown from the side view: tilted helix hairpin (H2, H3) of ATPTG11 mediates interactions with ATPTG5, ATPTG8 and ATPTG10, whereas ATPTG6 interacts with ATPTG8.

parallel to the membrane plane and mediates interactions between the dimers. H2$_{TG11}$ contacts with subunits ATPTG5, ATPTG8 and ATPTG10, whereas H3$_{TG11}$ interacts with ATPTG8 (Fig. 5b). Apart from the inward-facing residues, the ATPTG11 helix hairpin and interacting subunit segments are predominantly hydrophilic and solvent-exposed in the dimer.

For ATPTG9, we found that the four CX$_9$C motifs are conserved in mitochondriate Apicomplexa (Supplementary Fig. 10a, d). Likewise, we identified apicomplexan orthologs of ATPTG5 and ATPTG11, including candidate genes in *Plasmodium* with conserved cysteine pairs of the helix hairpins and key residues, including the N-terminal helix of ATPTG11 (Supplementary Fig. 10b, c, e, f). These data suggest that mitochondrial ATP synthase hexamers are a common feature of Apicomplexa.

**Hexamers specifically assemble into pentagonal pyramids in curved membrane regions.** To investigate if the mitochondrial ATP synthase hexamers occur in situ, we performed cryo-ET of isolated *T. gondii* mitochondrial membranes. Tomograms showed that the inner-membrane vesicles frequently displayed bulbous protrusions decorated with ATP synthase arrays (Fig. 6a, Supplementary Movie 2). Using subtomogram averaging, we then obtained a 20-Å resolution map of the ATP synthase dimer, which agrees well with the atomic model (Supplementary Fig. 11a, c). The analysis of the macromolecular arrangement of ATP synthase dimers in the membrane suggested that they are arranged into regular arrays with a hexamer as the repetitive unit, confirming the occurrence of this cyclic oligomeric form in situ (Fig. 6a).

Our data further show that these ATP synthase hexamers are arranged in larger arrays with icosahedral symmetry. The most frequently observed arrangement consists of ten ATP synthase

dimers arranged into five hexamer units, forming a pentagonal pyramid (Fig. 6a and Supplementary Movie 2). In the pyramid, the five inner dimers form a pentameric interface, with the C$_5$ symmetry axis centred on the apex of the vesicular protrusion (Fig. 5b). Each of the five inner dimers is shared by two neighbouring hexamer units (Fig. 5b).

To understand how the pentagonal pyramid induces the membrane curvature, we analysed cross-sections of the array (Fig. 6b, d, e). This showed that while the lipid bilayer within hexamers is near-planar (Supplementary Fig. 2e), neighbouring hexamers planes are related by a 45° angle (Fig. 6d). This is consistent with two times the 22° incline of each dimer with respect to the C$_3$ symmetry axis through the hexamer plane (Figs. 1c, d and 5b), indicating that the single-particle hexamer structure is consistent with the in situ pyramid assembly. Thus, membrane curvature is induced locally around the five inner dimers. Furthermore, hexamer planes are oriented by 40° with respect to the pentamer plane formed in the centre of the pyramid (Fig. 6a, b, e).

Fitting the dimer models into the pentagonal pyramid array suggested that no additional contacts are formed at the pentamer site. This indicates that the assembly of the pentagonal pyramids is fully explained by the contacts between the lumenal regions. This involves the same interactions as in the hexamer (Figs. 5 and 6c). Due to the C$_2$-symmetry of the dimer, each linker subunit is present in two copies, allowing the propagation of interactions (Fig. 6c), which results in the formation of a ~3.6-MDa array in the cristae lumen (Fig. 6a).

**Pentagonal pyramids are required for maintenance of native cristae architecture.** *Toxoplasma* tachyzoites are a model system to study mitochondrial functions in Apicomplexa, as they can be

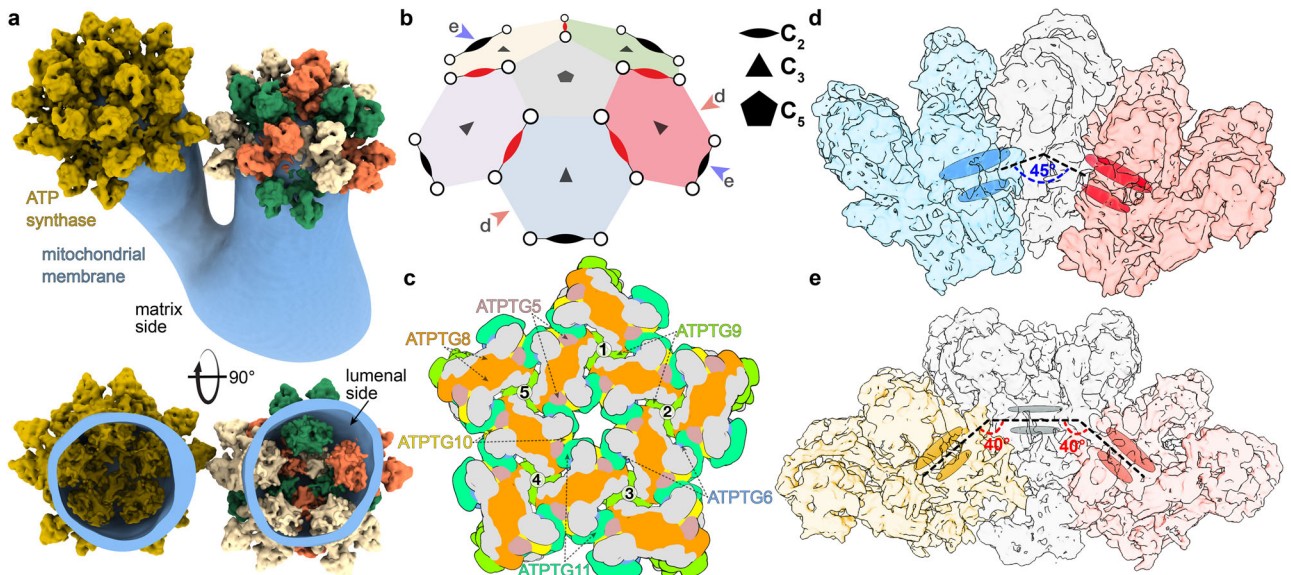

**Fig. 6 _T. gondii_ ATP synthase arranges into pentagonal pyramids with icosahedral symmetry to induce membrane curvature. a** Cryo-ET of the mitochondrial membranes (blue) and subtomogram averaging of dimers reveals their macromolecular arrangement into pentagonal pyramids held together by proteins in the lumen. **b** Schematic representation shows five hexamers (coloured) arranged around a $C_5$-axis. Red and blue arrows indicate cross-sections shown in the other panels. Inner and outer dimers are shown as red and black ellipses, respectively. **c** Schematic of interactions between luminal subunits involved in the assembly of the pentagonal pyramid. **d** Neighbouring hexamer planes (blue and red) are arranged at a ~45°-angle around the shared dimer (grey). **e** Cross section through the pentagonal ATP synthase pyramid showing two 40°-angles between hexamer (yellow, red) and pentamer (grey) planes.

cultured using alternative energy sources to oxidative phosphorylation, thereby enabling the mutation of genes encoding proteins involved in mitochondrial energy conversion[44]. To investigate the role of ATP synthase hexamers in maintaining native cristae architecture, we generated a knockout line of ATPTG11 (Supplementary Fig. 12a–c). Native gel electrophoresis confirmed that dimer assembly occurs in the absence of ATPTG11 (Supplementary Fig. 12d), and cryo-ET of mitochondrial membranes isolated from the ATPTG11-KO line revealed an altered organisation of the dimers in situ (Fig. 7, Supplementary Movie 3). Particularly, instead of forming pentagonal pyramids (Fig. 7a, c), ATP synthase dimers were found loosely arranged into disordered or row-like arrays along flat membrane regions (Fig. 7b, d). This demonstrates that ATPTG11 lumenal interfaces hold the pentagonal pyramids together (Fig. 6c). Visualization of the ATPTG11-KO mitochondrial membranes by cryo-ET revealed an elongated tubular shape (Fig. 7d), indicating that the formation of hexamers and pentagonal pyramids is critical for the maintenance of the bulbous cristae morphology in _T. gondii_. Thus, the ATPTG11-KO demonstrates the role of specific oligomer contacts in cristae architecture.

In addition, analysis of thin sections showed that ATPTG11-KO contains fewer cristae per mitochondrial area than the parental line, indicating an altered crista structure (Supplementary Fig. 13a–c). Flow cytometry using the potential-sensitive fluorescent dye JC-1 indicated that the mitochondria of ATPTG11-KO remain energized by a mitochondrial membrane potential, which is sensitive to the ionophore valinomycin, like the parental line (Supplementary Figs. 13d and 14). Fluorescence microscopy further revealed that the single large mitochondrion of both lines forms the characteristic lasso-shape, indicating that overall mitochondrial ultrastructure is not affected. Finally, we performed a growth competition assay where ATPTG11-KO parasites were grown in a mixed population with the parental line. Quantitative PCR of isolated genomic DNA (gDNA) during continued culturing showed that the relative abundance of the

ATPTG11-KO decreased significantly (Fig. 7e, f). These results indicate that the loss of ATP synthase oligomers and aberrant morphology are linked to impaired parasite fitness, when compared to the parental line.

In summary, we demonstrate that ATPTG11-KO selectively disrupts the formation of higher oligomers, while assembly of dimers appears unaffected. The resulting mild phenotype in cultured tachyzoites is in contrast to the strong growth defect that accompanies ATP synthase disassembly following loss of the indispensable core subunit-$b$[18]. This suggests that _T. gondii_ tachyzoites, which utilize both glycolysis[45] and oxidative phosphorylation[46] can compensate for the aberrant macromolecular organisation of ATP synthase, but not for the complete loss of its catalytic function.

**Pentagonal pyramid arrays shape unique cristae of _Toxoplasma_ mitochondria**. To confirm the occurrence of the pentagonal pyramids _in organello_, we performed cryo-ET of _T. gondii_ mitochondria with a translucent matrix. The analysis showed that cristae display bulbous morphology and are attached to the inner boundary membrane by circular cristae junctions (Fig. 8a). The bulbous cristae protrusions are capped by apical arrays of ATP synthases arranged into the pentagonal pyramids (Fig. 8b, Supplementary Fig. 15). Thus, the apicomplexan cristae morphology is in stark contrast to the coiled tubular cristae found in the related phylum of ciliates, which are shaped by helical ATP synthase dimer rows and connected via one crista junction at either end[47].

In addition, recent cryo-EM structures of a ciliate ATP synthase dimer and tetramer showed that although both structures share a small dimer angle and a large lumenal region[22], the two alveolate ATP synthases have diverged significantly and acquired lineage-specific subunits. Rather than forming hexamers, the ciliate-specific structural elements mediate dimer-dimer contacts that result in the formation of long helical ATP synthase

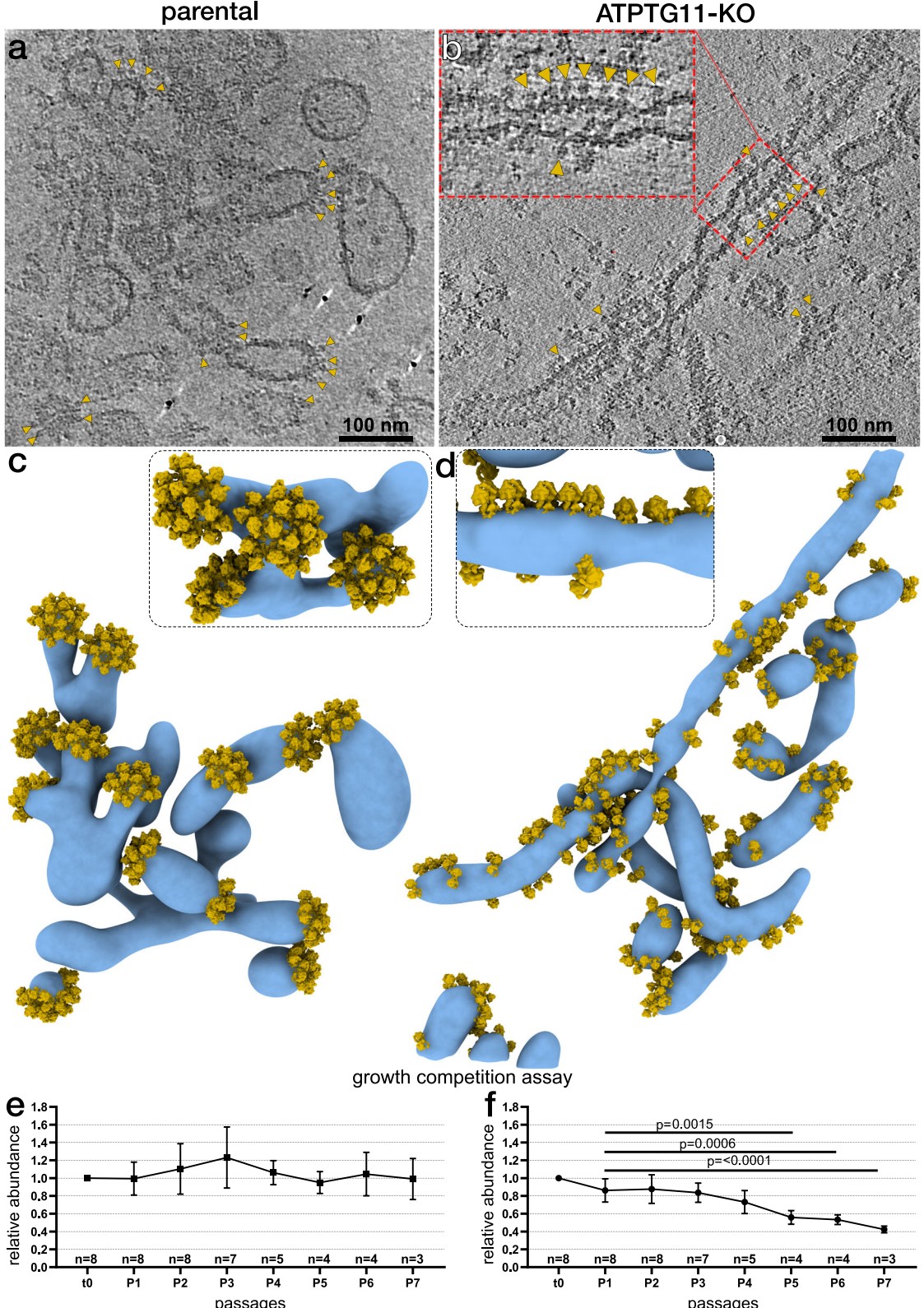

**Fig. 7 Pentagonal pyramids are required for maintenance of the native cristae morphology. a, b** Parental strain (**a**) and ATPTG11-KO (**b**) cross sections of tomograms of mitochondrial membranes decorated with ATP synthase (yellow arrows). **c, d** Segmentation of mitochondrial membranes (blue) with repositioned subtomogram averages of the dimers (yellow). Whereas the parental strain forms pentagonal pyramids that cap the bulbous membrane protrusions, hexamer and pyramid formation is disrupted in ATPTG11-KO, and ATP synthase dimers arrange in row-like or disordered arrays along elongated or tubular membranes. Close-up views show the pentagonal pyramid in the parental strain and row-like arrangements in the mutant strain. **e, f** Relative abundances of parental (**e**) and ATPTG11-KO (**f**) of the mixed-culture growth competition assay as determined by qPCR of total gDNA, normalized to t0. Each passage represents 3–8 biological replicates; error bars are SD; *p*-values were determined by one-way ANOVA followed by Dunnett's multiple comparisons test comparing each passage to P1.

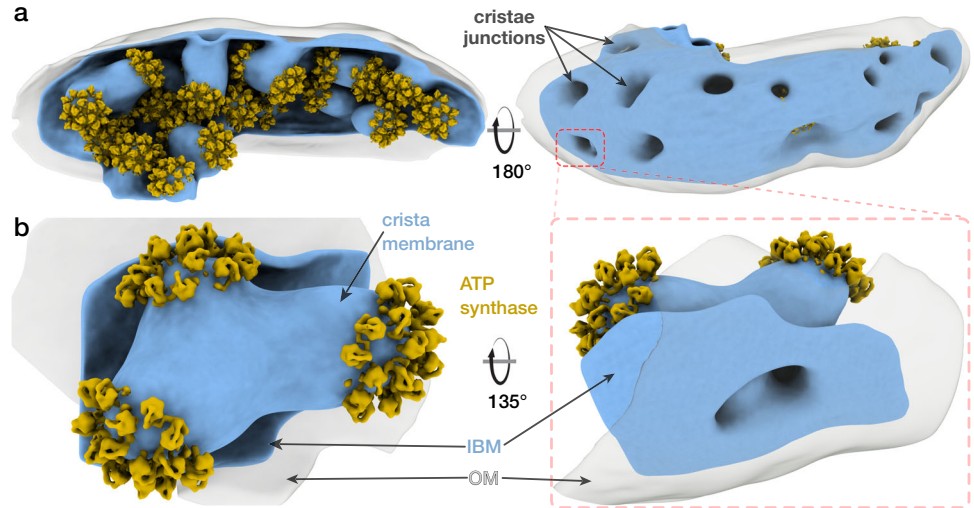

**Fig. 8 Pentagonal ATP synthase pyramid arrays decorate bulbous cristae in *T. gondii* mitochondria. a** Cryo-ET of a *T. gondii* tachyzoite mitochondrion (IBM, inner boundary membrane, blue; OM, outer mitochondrial membrane, grey) and subtomogram averaging of ATP synthase dimers (yellow). Cristae are connected to the inner boundary membrane via circular cristae junctions and decorated with pentagonal ATP synthase pyramids. **b** Close-up view of a crista membrane containing three bulbous protrusions, each decorated with an ATP synthase array containing ten ATP synthase dimers.

dimer rows and cristae tubulation. Thus, the superphylum of alveolates contains at least two different types of ATP synthase dimers and cristae morphologies, differentiating the free-living and parasitic protist phyla.

Our data suggest that in the organellar context, the exclusive localisation of the ATP synthase in the curved membrane regions will result in its segregation from the respiratory chain complexes residing in the flat cristae regions[6]. Such preferential localization of proton sinks has been suggested to generate a directional proton flow along a lateral proton gradient inside the cristae[48]. Together with the recent visualisation of cristae as high-potential compartments[12], these results suggest that assembly of a membrane-shaping ATP synthase oligomer drives its localisation to regions of high membrane potential, thus favouring ATP synthesis.

In summary, this work demonstrates that ATP synthase can be arranged in previously unseen high oligomeric arrays, which differ from the spontaneously assembled dimer rows, that were thought to be universal in all mitochondria[5–7,47,49,50]. We describe an organisational principle based on specific interactions between ATP synthase hexamers that are assembled into a pentagonal pyramid architecture. This results in the induction of local membrane curvature, which gives rise to the unique bulbous cristae morphology in Apicomplexa.

## Methods

**Cell culture and mitochondria isolation**. *T. gondii* RH tachyzoites were grown in Vero cells in DMEM supplemented with 10% (w/v) FBS, 2% (w/v) L-glutamine and 29.9 mM penicillin, 17.2 mM streptomycin at 37 °C with 5% (v/v) $CO_2$. For each mitochondrial preparation ~100 T150 flasks were harvested at >80% host-cell lysis and the media passed through 23G needles to fully lyse any remaining host cells. Parasites were pelleted by centrifugation at 1500 × *g*, 10 min, 4 °C, washed in PBS and then resuspended in buffer containing 210 mM mannitol, 70 mM sucrose, 50 mM HEPES-KOH pH 7.4, 1 mM EGTA, 5 mM EDTA, 10 mM KCl, 1 mM DTT to 5 × 10^8 cells/ml. Parasites were lysed by successive rounds of nitrogen cavitation (2500 PSI, 15 min incubation on ice) until >95% lysis (confirmed by light microscopy). After each round, the lysate was centrifuged at 1500 × *g*, 15 min, 4 °C, the supernatant was collected and the pellet resuspended in the same volume for further lysis.

The final combined lysate was centrifuged as before and the supernatant was spun again at 16,000 × *g*, 30 min, 4 °C. The resulting crude mitochondrial pellet was further purified on a discontinuous sucrose gradient in 20 mM HEPES-KOH pH 7.4, 2 mM EDTA, 15/23/32/60% (w/v) sucrose by centrifugation (103,745 × *g*, 1 h, 4 °C) in an SW41 rotor (Beckman Coulter) and enriched mitochondria were collected from the 32–60% (w/v) interface.

**Thin sectioning and conventional transmission electron microscopy**. Infected human foreskin fibroblasts (HFF) cultured in 6-cm dishes were washed gently with PBS. Light microscopy was used to ensure that the majority of extracellular parasites were removed. Host cells with intracellular parasites were gently scraped into 1.5 ml PBS and spun at 1500 × *g* for 10 min at RT. PBS was removed and cells were gently resuspended in freshly made fixation buffer (2.5% (v/v) glutaraldehyde, 4% (w/v) paraformaldehyde, in 0.1 M cacodylate buffer, pH 7.2), washed in 0.1 M cacodylate buffer, pH 7.2 and post-fixed in 1% (w/v) $OsO_4$, 1.25% (w/v) $K_4$[Fe(CN)$_6$] for 1 h on ice. After several washes in the same buffer, the samples were en bloc stained with 0.5% (w/v) uranyl acetate in water for 30 min. Afterwards, samples were washed with water, dehydrated in ascending acetone series and resin embedded. Ultrathin sections (~50 nm thick) were collected and imaged on a JEOL 1200 Transmission electron microscope (JEOL, Japan) operated at 80 kV. Obtained images (66 parental; 63 ATPTG11-KO) were analyzed with the Fiji software[51] and the number of individual cristae cross sections per mitochondrial area was calculated for the parental (80.8 ± 24.0 cristae/μm², mean, SD; *n* = 118) and ATPTG11-KO lines (59.6 ± 21.8 cristae/μm²; mean, SD; *n* = 103).

**ATPTG11 knockout line generation**. ChopChop tool (https://chopchop.cbu.uib.no/) was used to identify suitable gRNA near the start codon of the ATPTG11 gene (gRNA used: GATTGCGCACCATCTTGCAC). The gRNA was cloned under the U6 promoter (via BsaI restriction site) into a plasmid, which also encoded CAS9-GFP under the control of a TUB8 promoter[52] using the primers listed in Supplementary Table 3. A dihydrofolate reductase (DHFR) resistance cassette was amplified using the pDT7S4 plasmid as template[53] and using primers containing 50 bp of sequence homology to regions upstream and downstream of the ATPTG11 open reading frame. The plasmid and PCR product were co-transfected into an RH ΔHX Δku80 TATi parasite strain[53] and cassette integration was selected for with pyrimethamine for 8 days. The resulting parasite pool was cloned into 96-well plates with manual dilutions and 7 days later individual clones were tested by PCR for cassette integration using testing primers (Supplementary Fig. 12a, b).

**Real-time quantitative PCR**. To assess the expression level of ATPTG11 in the newly generated KO-line, ~5 × 10^6 freshly lysed parasites were filtered through 3-μm filters and collected by centrifugation. RNA was extracted from parasite pellets using the RNeasy kit (Qiagen) following manufacturer's instructions with the following modification: DNA was additionally on-column digested with 1 μl of amplification grade DNaseI (ThermoFisher Scientific) for 15 min at RT during step 4 of the manufacturer's protocol. Samples were reverse transcribed to cDNA using the High-Capacity RNA-to-cDNA kit (ThermoFisher Scientific) following manufacturer's protocol. Twenty nanograms of cDNA were then used in each qPCR reaction, which was set up with Power SYBR Green Master Mix (ThermoFisher Scientific) with 300 nM of each primer. All qPCR reactions were performed using a 7500 Real-Time PCR System (Applied Biosystems) using default temperature settings and performing a dissociation step after each run. Relative gene expression was determined using the $\Delta\Delta C_T$ method[54] using *T. gondii* catalase as an endogenous reference. The experiment was performed for ATPTG11-KO and the parental parasite line using primers against the unmodified ATPTG11 locus.

**Growth competition assay**. A confluent HFF monolayer was inoculated with ~1:1 ratio of ATPTG11-KO and parental parasites and incubated as described above. After complete host cell lysis, the collected parasites were mixed thoroughly, a new HFF dish was inoculated and the remaining parasites were filtered (3-μm pore size) and collected for gDNA extraction. gDNA was extracted using QiaGen DNeasy Blood & Tissue Kit. Power SYBR Green Master Mix, 300 nM of each primer and 10 ng of gDNA were used to perform quantitative PCR (7500 Real-Time PCR System using default settings with added dissociation step). The relative abundance of each parasite line was calculated relative to *T. gondii* catalase and normalized to the first collection point ($t_o$) using the $\Delta\Delta ct$ method[54] using the primers against the native locus and DHFR. Data were analysed using GraphPad Prism 8.4.3.

**Immunofluorescence assay and microscopy**. Parasites were inoculated on fresh HFFs on glass coverslips. After 1-day cells were fixed with 4% (w/v) paraformaldehyde. Cells were permeabilised and blocked with a solution in 2% bovine serum albumin and 0.2% (v/v) triton X-100 in PBS before incubation with primary antibodies (rabbit anti-TgMys[55], 1:1000, followed by secondary antibodies (Alexa Fluor Goat anti-Rabbit 594 Invitrogen #A-11012, 1:1000). Coverslips were mounted on slides with Fluoromount-G mounting media containing DAPI (Southern Biotech, 0100-20). Images were acquired via a DeltaVision Core microscope (Applied Precision) using a ×100 objective[55]. A total of seven representative images of ATPTG11-KO (containing individual vacuoles) and 10 images of parental parasites (containing 44 individual vacuoles) were obtained from two biologically independent repeats. Images were processed and deconvolved using the SoftWoRx (Glasgow, UK) and Fiji software[51].

**Flow cytometry analysis of membrane potential using JC-1**. Parasites grown in HFF were allowed to lyse the host cells. Collected parasites were filtered through a 3-μm filter and incubated in their growth media with 10 μM valinomycin for 30 min at 37 °C (as a depolarising control) or with an equal volume of DMSO, and then with 1.5 μM of JC-1 (5,5′,6,6′-Tetrachloro-1,1′,3,3′-tetra-ethylbenzimidazolocarbocyanine iodide, Thermo Fisher Scientific, stock 1.5 mM in DMSO) for 15 min at 37 °C before analysis with the CytoFLEX (Beckman Coulter Life Science), using an excitation wavelength of 488 nm and a 585/42 nm bandpass filter for detection of red fluorescence. 50,000 events per condition were collected and data were analysed using FlowJo (FlowJo LLC) to visualise the population of parasites with red fluorescent signal.

**Blue-native polyacrylamide gel electrophoresis and immunoblotting**. Whole parasites ($5 \times 10^6$) were mixed with 5 μl solubilisation buffer (750 mM aminocaproic acid, 50 mM Bis-Tris–HCl pH 7.0, 0.5 mM EDTA, 1% (w/v) dodecyl maltoside) and incubated on ice for 10 min. The resulting lysate was centrifuged at $18,000 \times g$ at 4 °C for 30 min. Sample buffer was added to the supernatant (NativePAGE™ 5% (w/v) G-250 Sample Additive and NativePAGE™ Sample Buffer (4X) (Invitrogen™), with a final concentration of Coomassie of 0.0625% (w/v)), resulting in a final concentration of 0.25% (w/v) dodecyl maltoside. NativePAGE™ Running Buffer (20X) and NativePAGE™ Cathode Buffer Additive (20X) (Invitrogen™) were mixed to reconstitute the anode, dark and light cathode buffers according to the manufacturer's instructions. Samples were loaded on 3-12% (v/v) Bis-Tris Gel (Novex- Life technologies) and 5 μl NativeMark™ (Invitrogen) was used as a molecular weight marker. Gels were run for 1 h at 80 V, 10 mA at 4 °C with dark cathode buffer, then for ~2 h at 200 V, 6 mA with light cathode buffer.

Proteins were transferred from the gel onto a methanol-soaked PVDF membrane (0.45 μm, Hybond™). Wet transfer in Towbin buffer (0.025 M Tris 0.192 M glycine 10% (v/v) methanol) was performed for 60 min at 100 V. The membrane was stained with Coomassie solution (50% methanol, 7% (v/v) acetic acid, and 0.05% (w/v) Coomassie R250 (Serva)) to visualise the molecular weight marker, and destained with 50% methanol, 7% acetic acid. Blots were labelled with primary rabbit anti-ATP-β (1:2000, Agrisera) coupled to secondary horseradish peroxidase (HRP) anti-rabbit (Promega) conjugated antibodies (1:10,000) and visualised using the Pierce ECL Western Blotting Substrate (Thermo Scientific).

**Purification of *T. gondii* ATP synthase dimers and hexamers**. Enriched mitochondria were lysed in a total volume of 34 ml buffer containing 25 mM HEPES/KOH pH 7.5, 25 mM KCl, 15 mM MgOAc₂, 2% (w/v) β-DDM, 2 mM DTT, 1 tablet EDTA-free Protease Inhibitor Cocktail for 2 h at 4 °C and the lysate was cleared by centrifugation at $30,000 \times g$, 20 min, 4 °C. The supernatant was layered on a sucrose cushion in buffer of 1 M sucrose, 25 mM HEPES/KOH pH 7.5, 25 mM KCl, 15 mM MgOAc₂, 1% β-DDM, 2 mM DTT, and centrifuged $230,759 \times g$, 4 h, 4 °C in a Ti70 rotor (Beckman Coulter). The resulting pellet was resuspended in 200 μl 25 mM HEPES/KOH pH 7.5, 25 mM KCl, 15 mM MgOAc₂, 2 mM DTT, 0.05% β-DDM, and gel filtrated over a Superose 6 Increase 3.2/300 column (GE Healthcare). Fractions corresponding to ATP synthase dimers were pooled and concentrated to 25 μl in a vivaspin500 filter (100-kDa MWCO). Purification of ATP synthase hexamers was performed similarly to that of dimers, but substituting β-DDM with identical concentrations of digitonin.

**Electron cryo-microscopy and data processing**. 3 μl ATP synthase dimer sample (~5 mg/ml) were applied to glow-discharged Quantifoil R1.2/1.3 Cu grids and

vitrified by plunge-freezing into liquid ethane after blotting for 3 s. The ATP synthase dimer was imaged on a Titan Krios operated at 300 kV at a magnification of 165 kx (0.85 Å/pixel) with a Quantum K2 camera (slit width 20 eV) at an exposure rate of 7.5 electrons/pixel/s with a 4-s exposure fractionated into 20 frames[56] using the EPU software (Thermo Fisher Scientific). A total of 4860 collected movies were motion-corrected and exposure-weighted using MotionCor2[57] and contrast transfer function (CTF) estimation was performed using Gctf[58]. Subsequent image processing was performed in RELION-3 (Supplementary Fig. 1)[59]. Bad images were removed manually by inspection in real- and Fourier-space. References for particle picking were generated from the data by Gaussian-blob picking and initial rounds of refinement and classification. Reference-based particle picking was performed using Gautomatch (developed by Dr Kai Zhang, MRC Laboratory of Molecular Biology, Cambridge, UK, http://www.mrc-lmb.cam.ac.uk/kzhang/Gautomatch) to pick 275,030 particles, which were subjected to reference-free two-dimensional (2D) classification, resulting in 214,085 particles for three-dimensional (3D) classification, from which 101,505 particles were selected. Masked refinements of the entire dimer and the membrane region with applied $C_2$-symmetry yielded maps of at 2.8 Å and 2.9 Å resolution, respectively. The pre-aligned particles were $C_2$-symmetry expanded and one monomeric unit was signal subtracted. Using masked 3D refinement, maps of the OSCP/$F_1$/c-ring, peripheral stalk and the rotor were obtained at 3.1 Å, 3.5 Å and 3.5 Å resolution, respectively. For final map generation, the original, rather than signal-subtracted particles were used. Focused classification of the $F_1$/c-ring and $IF_1$ binding regions did not reveal a presence of additional rotational states, or classes missing $IF_1$. Therefore, the inhibition by $IF_1$ is likely the result of the biochemical preparation.

*T. gondii* ATP synthase hexamers (19 mg/ml) were frozen as described above and imaged on a Titan Krios operated at 300 kV equipped with a K2 Summit detector and energy filter (20 keV slit width). Two datasets with a total of 7604 micrographs were acquired at a nominal magnification of 130 kx (1.05 Å/pixel) with a total exposure of 32 electrons/Å² over 6.5 s, fractionated into 20 frames. Initial picking references were generated from the data for reference-based particle picking using Gautomatch. Subsequent image processing in RELION-3 using 2D and 3D classifications (Supplementary Fig. 2) yielded a 4.8-Å resolution map of the hexamer membrane region from 4532 particles.

All final maps were generated from CTF-refined particles. All resolution estimates are according to Fourier shell correlations (FSC) that were calculated from independently refined half-maps using the 0.143-criterion with correction for the effect of the applied masks (Supplementary Figs. 1 and 2).

**Electron cryo-tomography and subtomogram averaging**. Crude *T. gondii* mitochondria pellets from either the parental or ATPTG11-KO strain were resuspended in an equal volume of buffer containing 20 mM HEPES-KOH pH 7.4, 2 mM EDTA, 250 mM sucrose and mixed in a 1:1 ratio with 5-nm colloidal gold solution (Sigma Aldrich) and vitrified as described above on glow-discharged Quantifoil R2/2 Cu grids. Tilt series were acquired on a Titan Krios operated at 300 kV with a Quantum K2 camera (slit width 20 eV) using serialEM[60] or the EPU software. Mitochondrial membranes were imaged at a nominal magnification of 64 kx (2.21 Å/pixel) and an exposure rate of 1.5 electrons/pixel/s with a 2-s exposure fractionated into four frames with tilt series acquired using the exposure-symmetric scheme[61] to ±60° tilt and a 3° tilt increment. Mitochondrial ghosts were imaged at a nominal magnification of 33 kx (4.27 Å/pixel) and an exposure rate of 11.5 electrons/pixel/s with a 3-s exposure fractionated into 3 frames. Bidirectional tilt series were acquired from −60° to 60° starting at 24° with a 2° tilt increment and a defocus range of −5 to −8 μm. Frames were motion-corrected and exposure-weighted using MotionCor2[57] and CTF estimation was performed using Gctf[58].

Tomographic reconstruction was performed in IMOD[62] using phaseflipping[63] and a binning factor 2. Tomograms were contrast enhanced using nonlinear anisotropic diffusion filtering[64] to facilitate manual particle picking of ATP synthases. Subtomogram averaging was performed in PEET[65]. Initial references were generated from the data by averaging after rotating subvolumes into a common orientation with respect to the membrane. Following initial rounds of subtomogram averaging, false-positive particles were removed based on a cross-correlation coefficient cut-off and manually by visual inspection of their orientation (e.g. removal of upside-down particles). Particles were then split into odd and even half-sets and aligned to independently updated references using $C_2$ symmetry using a mask around the ATP synthase dimer. A 20-Å resolution map was obtained from 139 ATP synthase dimers from one tomogram of mitochondrial membranes (Supplementary Figs. 11 and 15b), whereas a 22-Å resolution map of the ATPTG11-KO dimer was obtained from 269 particles from two tomograms. A 34-Å resolution map was obtained from 410 ATP synthase dimer particles from one tomogram of a *T. gondii* mitochondrion (Supplementary Fig. 15). Final maps were lowpass-filtered according to the 0.143-FSC criterion using RELION[66].

**Atomic model building and refinement**. Manual building of atomic models was performed in Coot[67]. $F_o$ subunits were built de novo using reconstructions of $F_o$ and peripheral stalk respectively (Supplementary Fig. 1). Built subunits were verified by BLAST searches against two libraries of putative *T. gondii* ATP synthase subunits[18,68]. Three $F_o$ subunits were identified by BLAST search using the built sequences against ToxoDB (toxodb.org) (Extended Data Table 1). OSCP/$F_1$/c-ring models were built using a homology model[69] of the yeast $F_1$/$c_{10}$-ring (PDB ID

3ZRY) [https://doi.org/10.2210/pdb3ZRY/pdb][70], whereas OSCP C-terminal domain and IF$_1$ were built de novo in an F$_1$/c-ring masked reconstruction. A total of two adenosine diphosphate molecules were resolved in three of the β-subunits, whereas three ATP molecules were resolved in the three alpha subunits. The database sequence the C-terminal helix of the α-subunit did not match the density map, but the manual building of the helix successfully identified the correct sequence for this part, corroborated by a single transcriptome study[71]. Real-space refinement of atomic models was performed in PHENIX[72] using secondary structure restraints and Rosetta[73]. Bound cardiolipins were unambiguously identified from their head group density. Other natively bound lipids were tentatively modelled as phosphatidyl choline or phosphatidyl ethanolamine, based on head group densities. To generate a composite model of the complete ATP synthase dimer, the atomic models of the membrane region, the OSCP/F$_1$/c-ring and the peripheral stalk were combined after rigid-body fitting into the consensus map of the dimer and refined in PHENIX using reference restraints. For the generation of an atomic model of the ATP synthase hexamer, individual models of ATP synthase dimer membrane region, peripheral stalk and F$_1$/c-ring were manually fitted into the hexamer reconstruction using Chimera[74]. The model fragments were combined into a single model file in Coot. The final hexamer model was real-space refined in PHENIX using the secondary structure and reference model restraints. Model statistics were calculated using MolProbity[75] and EMRinger[76]. To evaluate potential overfitting of the atomic models during refinement, the atomic coordinates of the refined models were randomly displaced by shifts up to 0.5 Å using 'Shake' in the CCPEM suite[77]. The shaken models were real-space refined using PHENIX against one-half map that had been reference-sharpened using Refmac[78]. Subsequently, FSC$_{work}$ and FSC$_{test}$ between the model and the two unfiltered half-maps, were calculated as described[78].

**Data analysis and visualisation.** Homology searches for *T. gondii* ATP synthase subunits across Myzozoans were performed in Eupath DB, EnsemblProstits and NCBI, using tBLASTn. The TGGT1_246540 gene (ATPTG1) is annotated on Toxo DB as cytochrome c1. This is due to the sequence containing two parts: the modelled ATPTG1 and a cytochrome c1. Because no part of the cytochrome c1 was found in the structure, homology searches were performed using the modeled ATPTG1 sequence only. This finding is consistent with a recent mass spectrometry study identifying peptides from the ATPTG1/cytochrome c1 gene in both *T. gondii* ATP synthase and complex III (https://doi.org/10.1101/2020.08.17.252163).

The luminal half-channel was traced as a void in the F$_o$-model, using the Caver plugin for PyMOL[79]. Calculation of subunit *a* surface electrostatics was done using APBS[80]. Images were rendered using PyMOL 2 (Schrödinger, LLC), Chimera[74] or ChimeraX[81]. The composite map of the ATP synthase dimer was generated in Chimera. This map was only used for visualisation, but not for atomic model refinement, where instead a consensus map was used. Prediction of cleavage sites of the mitochondrial matrix protease was performed using MitoFates[82]. Surface areas of subunit contacts were estimated using the PISA server for the dimerization interface or ChimeraX for the hexamer interface. Segmentation of membranes in tomographic volumes was performed manually in AMIRA (Thermo Scientific). Macromolecular arrangement of ATP synthase dimers was visualised by placing the subtomogram averages into the positions and orientations determined by subtomogram averaging using the clonevolume command in IMOD[83].

## Data availability
The atomic coordinates were deposited in the RCSB Protein Data Bank (PDB) under accession numbers 6TMG (membrane region), 6TMH (F1/c-ring), 6TMI (peripheral stalk), 6TMJ (rotor-stator), 6TMK (F$_1$F$_o$ dimer) and 6TML (hexamer). The cryo-EM maps have been deposited in the Electron Microscopy Data Bank (EMDB) under accession numbers EMD-10520 (membrane region), EMD-10521 (F$_1$/c-ring), EMD-10522 (peripheral stalk), EMD-10523 (rotor-stator), EMD-10524 (F$_1$F$_o$ dimer), EMD-10525 (hexamer), EMD-10526 (subtomogram average from mitochondrial membranes), EMD-11403 (subtomogram average from ATPTG11-KO mitochondrial membranes) and EMD-10527 (subtomogram average from mitochondria). Previously reported structural data includes accession numbers EMDB-0667, PDB 3ZRY, PDB 5X3P, PDB 2TRX, PDB 2LQL, PDB 6RD4, PDB 6B8H, PDB 6N2Y, PDB 6J5K, PDB 6J5I, PDB 6RD9, PDB 6CP6. Source data are provided with this paper.

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

## Acknowledgements

We acknowledge the ESRF beamline CM01 for provision of beam time for experiment MX-2107, and would like to thank especially Eaazhisai Kandiah for the excellent support. We thank the Astbury Biostructure Laboratory at the University of Leeds for tomography data collection, and especially Rebecca Thompson for her dedicated help. We also thank Michael Hall for data collection at the SciLifeLab National cryo-EM facility. We thank Leandro Lemgruber of

Glasgow Imaging Facility for his support and assistance in this work, members of Sheiner lab for help with parasite growth and harvesting, members of the Kuvin Center for the Study of Infectious and Tropical Diseases, Hebrew University of Jerusalem, for providing work space and support with flow cytometry work. This work was funded by the Swedish Foundation for Strategic Research (FFL15:0325), Ragnar Söderberg Foundation (M44/16), European Research Council (ERC-2018-StG-805230), Knut and Alice Wallenberg Foundation (2018.0080), BBSRC (BB/N003675/1), Wellcome Investigator award (217173/Z/19/Z). A.M. is supported by an EMBO Long-Term Fellowship (ALTF 260-2017). A.A. is supported by the EMBO Young Investigator Program. L.S. is a Royal Society of Edinburgh Personal Research Fellow. The SciLifeLab cryo-EM facility is funded by the Knut and Alice Wallenberg, Family Erling Persson, and Kempe foundations.

## Author contributions

A.M., L.S. and A.A. designed the project. A.M., P.F., A.L., J.O., L.S. and A.A. performed preparation of mitochondria from parasites. L.S. and J.O. generated the mutant line. A.M. performed protein purification and biochemical characterization, prepared cryo-EM grids, collected and processed EM data. A.M. and R.K.F. built, refined and validated the structures. A.M., R.K.F. and A.A. wrote the manuscript with help from L.S. All authors contributed to revising the manuscript.

## Funding

## Competing interests

The authors declare no competing interests.
