## [Peer Review File · Nature Communications]

REVIEWER COMMENTS

Reviewer #1 (Remarks to the Author):

The manuscript "ATP synthase hexamer assemblies shape cristae of *Toxoplasma mitochondria*" describes an unusual arrangement of ATP synthases of a human parasite *T.gondii* into hexamers by cryo electron microscopy. Structure of the ATP synthase dimer has been obtained by single particle cryo-EM at a resolution which allowed building an atomic model. This allowed characterizing the interactions within the dimer which contain a number of parasite-specific proteins conserved in Apicomplexa. The structure of hexamer showed that it contains 3 dimers. Such arrangement was found physiological by structural analysis by cryo electron tomography and subtomogram averaging. The authors suggest that the hexameric arrangement of ATP synthases organizes proton sinks in the mitochondria of apicomplexan parasites.

The results are very interesting to the wide structural biology and microbiology community, the experiments are excellently executed and the manuscript is clearly written with the appropriate credit given. I have a few comments that would further improve the manuscript to make it an excellent contribution to Nature Communications.

Major comments:

1. Page 5 Lines 37 and later: The fact that all the ATP synthases had the IF1 inhibitor bound is highly unusual and requires further investigation or comment. Could this just be the most thermodynamically stable conformation which is favoured by the classification procedure where ~214 k particles after 2D classification got reduced to 100k after 3D classification? Are there classes with lower resolution that do not have the inhibitor? If classes without the inhibitor cannot be identified, does it mean that all ATP synthases in *Toxoplasma gondii* are mostly inhibited in vivo?

2. Page 8: Hexamerization - the reviewer would like to know the energy contribution of the interactions between the dimers within a hexamer. What is the surface area of the interfaces if this is only hydrophobic and are there any suggested hydrogen bonds? Are there any conformational differences between the dimers alone and the dimers arranged in a hexamer? Finally, the authors calculate the angles between the neighbouring hexamer planes in situ, the same measurement could be calculated for the angles between the planes of the dimers in purified hexamers for comparison.

3. The most important question - what is the physiological effect of the ATPTG11-KO on the parasite? Does it grow slower in culture / deficient in energy production / has any further defects? Further characterization of the mutant should be performed and presented in the updated version of the manuscript.

Minor comment:

4. I suggest changing the order of the sections for the alpha subunit description and about the salt bridge formation in presence of the inhibitor (pages 5 and 6), this could make the flow of the paper smoother.

Reviewer #2 (Remarks to the Author):

The cryo-EM study by Mühleip and coworkers presents near-atomic resolution structural models for dimeric and hexameric mitochondrial ATP synthase from the parasite *Toxoplasma gondii*. *T. gondii* is closely related to the malaria parasite *Plasmodium falciparum*, and high-resolution structures of the

parasites' peculiar ATP synthase complexes may lead to therapeutics as ATP synthase is essential for the survival of the parasites. It was previously known that *T. gondii* mitochondrial inner membrane cristae have a morphology distinct from yeast or mammalian mitochondria, and the authors wished to know whether the oligomeric state of ATP synthase may be responsible for this difference. Using the mild detergent digitonin, they were able to isolate hexameric ATP synthase oligomers that are formed from three copies of the dimeric complex. The authors then complemented the high-resolution single-particle study by using cryo electron tomography to show that hexameric ATP synthase complexes are indeed present in *T. gondii* mitochondrial inner membranes, with the hexamers assembling into fivefold symmetric higher oligomers positioned at the tips of tubular cristae structures. From the high-resolution structural models, the authors identify an apicomplexan specific subunit (ATPTG11) that, when knocked out, leads to the disruption of the hexameric and higher oligomeric assemblies and to elongated tubular cristae structures. Another interesting (but not necessarily novel aspect) is that subunit 'a' is truncated, and that its gene has been moved from the mitochondrial genome to the nucleus. The truncation appears to decrease overall hydrophobicity, which may be a requirement for efficient mitochondrial import.

In general, the study is of high-quality and conducted with attention to detail (the comparison to well characterized and more familiar dimeric ATP synthase complexes from bacteria, yeast, and animal mitochondria is interesting and helpful). On the other hand, the study does have a significant component that is somewhat descriptive, which is obviously difficult to avoid due to the large number of apicomplexan specific subunits that are unfamiliar, and whose function is poorly understood. And there are instances where the authors make statements (e.g. distances between side chains, nature of lipid molecules) that are questionable given the still limited resolutions of some of the maps. The authors, however, make a significant effort to provide additional insight from electron tomography and mutagenesis experiments.

A few points to consider:

- Fig. 3B and Fig. S7E: The authors model the side chains of cGlu150 and aArg166 with a short distance of 2.3 Å, however from the view in Fig. 3B, there appears to be little or no density for Glu150's side chain (hard to see in the figure). Given the resolution of the map used for this modeling (3.5 Å), how confident can the authors be that the two side chains are modeled correctly, and/or that the side chains actually adopt a single rotamer? Moreover, they point to differences in the modeling of related ATP synthases (Fig. S7E), but they show no density for the related enzymes. If they want to make the point that the salt bridge between *T. gondii* cGlu150 and aArg166 is genuine and distinct from the arrangement of the c-ring:a interface of related enzymes, they should provide a clear illustration of the densities of all involved residues, including the ones for the related enzyme complexes.
- The statement that "...water is likely excluded from the region surrounding Arg166." seems highly speculative given the resolution of the intact membrane subcomplex map. Breaking the salt bridge in absence of water would be energetically highly unfavorable.
- Fig. S3B: It is hard to see whether the "membrane region" contains the c-ring or not. If not, maybe "membrane stator subcomplex" may be more accurate?
- Fig. S5: A few selected lipids should be shown in density to support the authors' assurance that lipids could be identified based on head group density (page 16).
- Fig. 2C: Change "blue" to "cyan" in the legend?

Reviewer #3 (Remarks to the Author):

Muhleip et al., presents a single particle density map of the mitochondrial ATP synthase dimer from

the parasite *Toxoplasma*. They observe that the ATP synthase dimer forms a trimer and that the interaction between dimers is caused by the *T. gondii* specific subunit ATPTG11. The substantiate their claim by generating a ATPTG11 mutant and observe that the trimer of dimers has disappeared. The work is well executed and well present. I only have a few minor comments:

line 43: "do" needs to be changed to "to"

line 65: instead of "resulted in 2.8-3.5 Å resolution maps" please change to: "resulted in density maps of the membrane region, the OSCP... and the peripheral stalk range in resolution from 2.8-3.5Å"

Line 88-92 and Fig 2A,B. I can't find ATPTG11 in Fig2B. Also I would like to clearly see how ATPTG11 plugs the c-ring cavity.

Line 31 page 5, add "An" to last sentence to read. An additional 15 cardiolipin.

Line 59 page 8, last sentence: H2TG11 contacta with..

Fig 5 is not clear. Where are the dimers in this fig? Does one dimer contribute to two hexamers? How do the hexamers shown in the single particle map fit into the tomographic maps? Is one hexamer pseudo? Is ATPTG11 found on both sides of the dimer and thus can participate in two trimer of dimer organisations?

Also, with the resolution obtained, the ATP synthases do not form hexamers but trimers of dimers. The symmetry is C3 not C6. Thus the title and wording in the manuscript needs to be changed to reflect this distinction.

Supplementary figure 4,

What is the difference between ATPTG1 and ATPTG15? Is there one copy per dimer or one copy per monomer. From SupFig4a, there doesn't seem to be any difference.

Reviewer #4 (Remarks to the Author):

There is considerable diversity in the morphology of cristae in eukaryotes. Apicomplexan parasites, such as the study species *Toxoplasma gondii*, have tubular cristae that differ in structure from that seen in mammalian cells. The mitochondrial ATP synthase complex is known to play a central role in modulating the shape of cristae. In this manuscript, Mühleip and colleagues establish the structure and structural arrangements of ATP synthase from *T. gondii*. The authors elucidate numerous divergent features of ATP synthase from *T. gondii* compared to the well-characterised ATP synthase from mammals and yeast. These include a novel Fo subdomain that constitutes the dimerization interface, a truncated and less hydrophobic subunit-a that is part of a divergent proton translocation mechanism, and an extension of the subunit-b in the peripheral stalk that appears to take the place of the missing subunit-h of the complex. Most strikingly of all are the hexameric assemblages of the ATP synthase complex in these parasites. The authors present some evidence that these assemblages contribute to the tubular shape of mitochondrial cristae, and they demonstrate that the higher order arrangements of the complex are dependent on the presence of subunits that are restricted to apicomplexan parasites. The study provides the first high resolution structural analysis of mitochondrial ATP synthase from an apicomplexan parasite, and sets the stage for future studies to characterise the divergent features of this complex. The manuscript is well written, and the structural images and movies are quite stunning. My comments are, for the most part, minor and can be addressed by the authors with minimal additional experimentation. I am not expert in structural determination, so my comments focus on aspects of the paper that explore the role of ATP synthase in mitochondrial biology of *T. gondii*.

Major comments

1. A lot of interesting data that is pertinent to the story is buried in supplementary data. Unless Nature Communications has a limit on figure numbers (which I don't think it does), I strongly suggest moving some parts of the supplementary figures to the main body of the text. In particular, data from the ATPTG11 knockout (parts of Supplementary Figures 9 and 10 and Movie S3), which provide an experimental test of hypotheses generated from the structural data, should be included in the main body of the text. Other aspects of the supplementary data that could be moved to the main body of the text include parts of Supplementary Figure 4, which highlight the apicomplexan-specific proteins at the dimer interface of the complex.

2. Regarding the studies on the ATPTG11 knockout, the data clearly show that loss of ATPTG11 impairs assembly of the ATP synthase multimeric arrangements. However, I don't get a strong sense of the effects that this has on cristae architecture. The manuscript would be strengthened by data that quantify, or better describe, the changes to cristae architecture that have occurred (e.g. measuring changes in length of the cristae). Also, the authors should provide additional data on whether loss of ATPTG11 leads to any defects in parasite proliferation or mitochondrial ATP synthesis, and comment on what these data mean for the role of cristae architecture/higher order assemblages of ATP synthase for parasite proliferation.

Minor comments

3. General comment on the introduction: a brief introduction on the different domains of ATP synthase (Fo, F1, rotor, stator, central stalk, peripheral stalk, etc) would help non-expert readers understand some of the later discussion.

4. Page 2, Line 43. "to" not "do" (typo)

5. Page 2, Line 48. "differs substantially in its cristae morphology" from what? It reads like the comparison is to Plasmodium, whereas I think the authors mean the 'bulbous'/tubular cristae that occur in apicomplexans compared to the flattened/lamellar cristae seen in yeast or mammalian cells (as noted in the subsequent sentence). Suggest rewording.

6. Page 2, Line 49. "bulbous cristae". Is the term bulbous used beyond the Ferguson study that is cited? The term "tubular" is more common in the literature around cristae morphology in eukaryotes. Are these "bulbous" cristae obviously different from, say, the tubular cristae in Plasmodium or ciliates to warrant a separate term?

7. Page 2, Lines 50-51. "Loss of ATP synthase and the resulting defect in cristae abundance leads to death of the *T. gondii* ..." – the cited studies indicate that ATP synthase is important for these stages in the parasites, and the Huet study depicts changes in mitochondrial morphology upon ATP synthase loss. But neither study establishes that the growth defects result from the defects in cristae abundance as the text implies. The authors should reword this to be clearer about what these studies do, and do not, show.

8. Page 3, Line 70. I'm curious about the reasoning behind the "ATPTG1-17" nomenclature. What does the TG stand for? *Toxoplasma gondii*? The authors could specify this.

9. Page 3, Line 70 and elsewhere: "apicomplexan-conserved" subunits. The authors could include additional details that provide evidence these subunits are conserved in apicomplexans (beyond the alignments shown in Supplementary Figure 8). Are they conserved across the apicomplexan lineage? And in other myzozoans? The answers may provide some insights into, for example, the likelihood that the newly identified Fo subcomplex is conserved across these lineages. Perhaps an addition to Table S2 could be used to clarify this?

10. Page 3, Lines 89-90. "This is mediated by a short N-terminal hydrophobic helix of ATPTG11, which is sequence-conserved in Apicomplexa (Fig. 2A,B)." Can the authors annotate ATPTG11 on the main part of Fig. 2A and in Fig 2B? I think this is the blue-green coloured subunit, but this is not obvious.

11. Page 5, Line 20. Where the unusual dimer architecture of the *T. gondii* ATP synthase is first mentioned, authors could consider briefly describing what the conventional dimerisation interface is comprised of, so that the reader has a better appreciation for the novelty of the data.

12. Page 5, Line 37. The authors state that the inhibitor protein of ATPase activity, IF1, contributes to the Fo dimer interface. Does this mean that dimerisation depends on IF1, or that ATP synthase can dimerise in the absence of IF1, but that IF1 inhibits ATP synthase by binding along the dimer interface to lock the two F1 regions into the ADP bound state? Or is it not possible to make any functional conclusions about the role of IF1 from the data? It would help readers understand the importance of these data by placing this into some more context.

13. Page 6, Lines 67-68. "A similar observation for different mitochondrial membrane proteins has been proposed to enable mitochondrial protein targeting following gene transfer to the nucleus (24, 25)." Also proposed for the Cox2a and 2b sequences of apicomplexan parasites – PMID 12843377.

14. Page 8, Line 59. "contacta with" – "contacts"?

15. Legend, supplementary Figure 8. What is AK88 (line 75) in the ATPTG9 alignment?

16. Movie S2. The movies are stunning, and a strong addition to the manuscript. On my version of this movie, the 'legend' depicting hexamers and pentamers flashes up only very briefly – I think it would be helpful for this to appear for a longer time so they can be viewed more easily.

17. Page 11, Line 20. "...enabling the generation of mitochondrial mutants that are otherwise lethal". The knockout in the BCKDH-E1a subunit to which the authors refer can still proliferate in glucose-free medium (albeit more slowly than in complete medium). I don't think it accurately reflects the literature to say that mitochondrial mutants are "otherwise lethal". It would be more accurate to state that parasites "... can be cultured using alternative energy sources to OxPhos, thereby enabling the mutation of genes encoding proteins involved mitochondrial energy metabolism."

18. As a follow-up point, the authors don't show that ATPTG11 knockout leads to defects in ATP synthesis in the mitochondrion, so the link to OxPhos that "allowed us to generate a knockout line of ATPTG11" is not established. In fact, a genome-wide CRISPR screen suggests that ATPTG11 is dispensable for tachyzoite proliferation (PMID 27594426), which will have indicated to the authors that this gene is unlikely to be essential for parasite proliferation or ATP synthesis before they went to knock this out. I suggest rewording this section. I don't think the authors need to justify this experiment by linking ATPTG11 to OxPhos – the rationale to study the role of ATPTG11 in maintaining cristae architecture is sufficient.

19. Supplementary Figure 10D. The authors should indicate where the ATP synthase dimer occurs on the western blot. I believe this should be the ~1048 MDa complex, while readers might think this is the ~720 kDa complex, since a monomeric band is more visible in the KO than in the parental strain.

20. Page 33, Supplementary Figure 9, Lines 91-92. "(I-J) thin sections of *T. gondii* with mitochondria outlined (red dashed lines), showing cristae morphology. " The authors should indicate in the figure legend which image is of WT mitochondria and which is of ATPTG11 mitochondria.

21. Page 12, Line 88. The sgRNA listed here appears to include the PAM site, which is not a component of the actual sgRNA.

22. Page 13, Lines 17-18. "The supernatant was added sample buffer" – "Sample buffer was added to the supernatant" perhaps?

We thank the reviewers for kindly providing a thorough assessment of our data and constructive comments that helped to improve the manuscript. The reviewers brought up a number of important points regarding the data presentation, details of the analysis, technical aspects of cryo-EM, and conclusions. Each one of the points has been addressed, as specified in the point-by-point response below.

To illustrate the aspects of the additionally performed structural analysis and functional characterisation, we added the following figures or panels to the revised version of the manuscript:

- two panels to Figure 1, showing the canonical and apicomplexa-specific structural components in the dimer, and the atomic model of the hexamer (Reviewer #4)
- panel to Figure 2 illustrating a cross section through the F_o region of the map to visualize how ATPTG11 plugs the c-ring (Reviewers # 2 and 3)
- improved Figure 6a,b highlighting individual dimers in the pentagonal pyramid (reviewer #3)
- two panels to Figure 7 illustrating the relative abundances of parental and KO strain in a competitive growth assay (Reviewers # 1 and 4)
- Supplementary Figure 4: phylogenetic analysis of ATP synthase subunits in Myzozoans (Reviewer #4)
- Supplementary Figure 11: fit of the dimer models in the subtomogram averages (Reviewer # 3)
- Supplementary Figure 13 describing characterisation of the ATPTG11-KO line in comparison with the WT (Reviewers #1 and #4).

In addition, a number of figures have been rearranged, and multiple formulations improved throughout the text, according to the guideline provided by the reviewers. Below is the detailed point-by-point response.

Reviewer #1 (Remarks to the Author):

The manuscript “ATP synthase hexamer assemblies shape cristae of *Toxoplasma* mitochondria” describes an unusual arrangement of ATP synthases of a human parasite *T.gondii* into hexamers by cryo electron microscopy. Structure of the ATP synthase dimer has been obtained by single particle cryo-EM at a resolution which allowed building an atomic model. This allowed characterizing the interactions within the dimer which contain a number of parasite-specific proteins conserved in Apicomplexa. The structure of hexamer showed that it contains 3 dimers. Such arrangement was found physiological by structural analysis by cryo electron tomography and subtomogram averaging. The authors suggest that the hexameric arrangement of ATP synthases organizes proton sinks in the mitochondria of apicomplexan parasites.

The results are very interesting to the wide structural biology and microbiology community, the experiments are excellently executed and the manuscript is clearly written with the appropriate credit given. I have a few comments that would further improve the manuscript to make it an excellent contribution to Nature Communications.

Major comments:

1. Page 5 Lines 37 and later: The fact that all the ATP synthases had the IF₁ inhibitor bound is highly unusual and requires further investigation or comment. Could this just be the most thermodynamically stable conformation which is favoured by the classification procedure where ~214 k particles after 2D classification got reduced to 100k after 3D classification? Are there classes with lower resolution that do not have the inhibitor?

To follow the reviewer's suggestion to identify potential classes lacking IF₁, we performed subclassification of F₁/c-ring complexes. However, no classes lacking IF₁ or additional rotational states, which would be expected to result from the absence of the inhibitor, were identified. This is now stated in Methods (page 20, lines 574-576). In previous studies, high-occupancy binding of IF₁ was reported for other types of mitochondrial ATP synthase (PMID: 31197009; PMID: 31738165).

If classes without the inhibitor cannot be identified, does it mean that all ATP synthases in *Toxoplasma gondii* are mostly inhibited in vivo?

Since *T. gondii* tachyzoites have been reported to possess energized mitochondria, which can drive ATP synthesis (suppl. Fig.13c, PMID: 10806088) and oxidative phosphorylation is essential for maintaining their ATP levels (PMID: 19286986), the inhibition by IF₁ is likely the result of the biochemical preparation. We now added this information (page 20 line 558-559).

2. Page 8: Hexamerization - the reviewer would like to know the energy contribution of the interactions between the dimers within a hexamer. What is the surface area of the interfaces if this is only hydrophobic and are there any suggested hydrogen bonds?

As requested, we calculated the surface area and added to the text together with the analysis of the interactions on page 9, lines 268-289. Each of the three dimer-dimer interfaces in the hexamer contributes a buried area of 1211 Å². Although the 4.8-Å map of the hexamer membrane region does not warrant interpretation of individual interactions at the side chain level or reliable calculation of energy contribution, we now state the hydrophilic character of the interacting structural elements (page 9, lines 276-277, 282-283), which is consistent with their luminal location (Fig 5), and that these elements are solvent-exposed in the dimer structure.

Are there any conformational differences between the dimers alone and the dimers arranged in a hexamer?

No significant conformational differences between the dimers alone and those arranged in a hexamer have been observed, and we added this information on page 4, line 103-105. We also added a reference to Supplementary Movie 1, where this aspect is further clarified.

Finally, the authors calculate the angles between the neighbouring hexamer planes in situ, the same measurement could be calculated for the angles between the planes of the dimers in purified hexamers for comparison.

We added this information on page 11, lines 316-320. The requested angle (dimer tilt against the hexamer plane) was originally measured as 22° (page 4, line 105-107), which, as expected purely by geometry, is almost exactly half the angle between two hexamer planes, that are connected by a shared dimer (thus $2 \times 22^\circ$, Figure 6b,d).

3. The most important question - what is the physiological effect of the ATPTG11-KO on the parasite? Does it grow slower in culture / deficient in energy production / has any further defects? Further characterization of the mutant should be performed and presented in the updated version of the manuscript.

We performed a competitive mixed growth assay, in which we infected the obligatory host cells with a mixed population of parental and ATPTG11-KO tachyzoites. Within a few passages, a significantly decreased abundance of the mutant parasite line was detected, indicating a decreased parasite fitness, which is now shown in Figure 7e. The information has been added on page 12, lines 364-368.

Minor comment:

4. I suggest changing the order of the sections for the alpha subunit description and about the salt bridge formation in presence of the inhibitor (pages 5 and 6), this could make the flow of the paper smoother.

We appreciate the suggestion and considered implementing it, however since the description of the salt bridge formation relies on the subunit-a structure, it would need to follow the former. To address the comment and make the flow of the paper smoother, we revised the text on page 7 that now clarifies the relationship between the sections.

Reviewer #2 (Remarks to the Author):

The cryo-EM study by Mühleip and coworkers presents near-atomic resolution structural models for dimeric and hexameric mitochondrial ATP synthase from the parasite *Toxoplasma gondii*. *T. gondii* is closely related to the malaria parasite *Plasmodium falciparum*, and high-resolution structures of the parasites' peculiar ATP synthase complexes may lead to therapeutics as ATP synthase is essential for the survival of the parasites. It was previously known that *T. gondii* mitochondrial inner membrane cristae have a morphology distinct from yeast or mammalian mitochondria, and the authors wished to know whether the oligomeric state of ATP synthase may be responsible for this difference. Using the mild detergent digitonin, they were able to isolate hexameric ATP synthase oligomers that are formed from three copies of the dimeric complex. The authors then complemented the high-resolution single-particle study by using cryo electron tomography to show that hexameric ATP synthase complexes

are indeed present in *T. gondii* mitochondrial inner membranes, with the hexamers assembling into fivefold symmetric higher oligomers positioned at

the tips of tubular cristae structures. From the high-resolution structural models, the authors identify an apicomplexan specific subunit (ATPTG11) that, when knocked out, leads to the disruption of the hexameric and higher oligomeric assemblies and to elongated tubular cristae structures. Another interesting (but not necessarily novel aspect) is that subunit 'a' is truncated, and that its gene has been moved from the mitochondrial genome to the nucleus. The truncation appears to decrease overall hydrophobicity, which may be a requirement for efficient mitochondrial import.

In general, the study is of high-quality and conducted with attention to detail (the comparison to well characterized and more familiar dimeric ATP synthase complexes from bacteria, yeast, and animal mitochondria is interesting and helpful). On the other hand, the study does have a significant component that is somewhat descriptive, which is obviously difficult to avoid due to the large number of apicomplexan specific subunits that are unfamiliar, and whose function is poorly understood. And there are instances where the authors make statements (e.g. distances between side chains, nature of lipid molecules) that are questionable given the still limited resolutions of some of the maps. The authors, however, make a significant effort to provide additional insight from electron tomography and mutagenesis experiments.

A few points to consider:

- Fig. 3B and Fig. S7E: The authors model the side chains of cGlu150 and aArg166 with a short distance of 2.3 Å, however from the view in Fig. 3B, there appears to be little or no density for Glu150's side chain (hard to see in the figure). Given the resolution of the map used for this modeling (3.5 Å), how confident can the authors be that the two side chains are modeled correctly, and/or that the side chains actually adopt a single rotamer? Moreover, they point to differences in the modeling of related ATP synthases (Fig. S7E), but they show no density for the related enzymes. If they want to make the point that the salt bridge between *T. gondii* cGlu150 and aArg166 is genuine and distinct from the arrangement of the c-ring: a interface of related enzymes, they should provide a clear illustration of the densities of all involved residues, including the ones for the related enzyme complexes.

- The statement that "...water is likely excluded from the region surrounding Arg166." seems highly speculative given the resolution of the intact membrane subcomplex map. Breaking the salt bridge in absence of water would be energetically highly unfavorable.

As requested, we have added map densities to supplementary Fig. 8e. We now clarify that like most, if not all available mitochondrial ATP synthase structures, our cryo-EM map does not display unambiguous density for the Glu150 (page 7, lines 200-202), likely due to the sensitivity of carboxyl groups to irradiation damage (see PMID: 33033219). We now emphasized that the rotamer placement relies on prior knowledge from X-ray structures, which show this residue to adopt one of two observed rotamers. We further reformulated the interpretation of the observed hydrophobic/aromatic surrounding of the Arg/Glu pair in *T. gondii* as being consistent with a direct salt bridge, rather than a water-mediated interaction.

- Fig. S3B: It is hard to see whether the “membrane region” contains the c-ring or not. If not, maybe “membrane stator subcomplex” may be more accurate?

We added a label with an arrow to the c-ring position to this figure. Furthermore, the new Fig 2c shows that this map contains clear c-ring density, which is why we prefer to keep the original naming.

- Fig. S5: A few selected lipids should be shown in density to support the authors' assurance that lipids could be identified based on head group density (page 16).

We added a panel with the density for each of the three lipid types, demonstrating that the assigned lipid identity is consistent with both the cryo-EM map and the coordinating side chains. This is now Supplementary Figure 6b.

- Fig. 2C: Change “blue” to “cyan” in the legend?

We have changed the description to light blue.

Reviewer #3 (Remarks to the Author):

Muhleip et al., presents a single particle density map of the mitochondrial ATP synthase dimer from the parasite *Toxoplasma*. They observe that the ATP synthase dimer forms a trimer and that the interaction between dimers is caused by the *T. gondii* specific subunit ATPTG11. They substantiate their claim by generating a ATPTG11 mutant and observe that the trimer of dimers has disappeared. The work is well executed and well present. I only have a few minor comments:

line 43: "do" needs to be changed to "to"

Implemented.

line 65: instead of "resulted in 2.8-3.5 Å resolution maps" please change to: "resulted in density maps of the membrane region, the OSCP... and the peripheral stalk range in resolution from 2.8-3.5Å"

Implemented.

Line 88-92 and Fig 2A,B. I can't find ATPTG11 in Fig2B. Also I would like to clearly see how ATPTG11 plugs the c-ring cavity.

We added a label for ATPTG11 in Figure 2b, where it can be seen inside the c-ring. We also expanded the description of how ATPTG11 plugs the c-ring on the border of the detergent belt, highlighting the amphipathic character of H1_{TG11}, which forms the c-ring plug with its hydrophobic residues (pages 3-4, lines 94-100). Finally, we added panel Figure 2c, showing a map cross-section with ATPTG11 extending from the luminal region of the stator to contact the inside of the c-ring rotor, which is embedded in the detergent belt.

Line 31 page 5, add "An" to last sentence to read. An additional 15 cardiolipin.

Added.

Line 59 page 8, last sentence: H2TG11 contacta with..

Corrected.

Fig 5 is not clear. Where are the dimers in this fig? Does one dimer contribute to two hexamers? How do the hexamers shown in the single particle map fit into the tomographic maps? Is one hexamer pseudo? Is ATPTG11 found on both sides of the dimer and thus can participate in two trimer of dimer organisations?

We clarified the figure (now Fig 6a) by colouring the right pentagonal pyramid to show individual dimer units within the assembly. We furthermore clarify that the pyramid consisting of 10 ATP synthase dimers contains 5 inner dimers (red ellipses in Figure 6b), each of which is shared between two neighboring hexamer planes. This is indicated in Figure 6d (white dimer) and explained in more detail in the text (page 11, lines 313-314, Figure 6 legend).

Because we averaged and repositioned the dimeric unit in subtomogram averaging, it cannot be compared directly to the single-particle hexamer. To show consistency of cryo-EM and cryo-ET data, we now added the fit of the dimer model in the subtomogram averages in Supplementary Fig 11b.

Also, with the resolution obtained, the ATP synthases do not form hexamers but trimers of dimers. The symmetry is C₃ not C₆. Thus the title and wording in the manuscript needs to be changed to reflect this distinction.

We have now included an additional emphasis on page 9, line 267-270 and labels in Figure 1d, stating that the hexameric assembly is formed as a trimer of dimers. The C₃ symmetry is further indicated in Figure 1c. Hexamer is a suitable description, as the mammalian counterpart, which is a dimer of dimers (C₂ symmetry, not C₄), has been described as a tetramer (PMID: 31197009; PMID: 32900941).

Supplementary figure 4, What is the difference between ATPTG1 and ATPTG15? Is there one copy per dimer or one copy per monomer. From SupFig4a, there doesn't seem to be any difference.

We regret that these two subunits were shown in similar colors. To clarify, we added labels to the figure (new Supplementary Fig 5a), indicating each subunit copy individually.

Reviewer #4 (Remarks to the Author):

There is considerable diversity in the morphology of cristae in eukaryotes. Apicomplexan parasites, such as the study species *Toxoplasma gondii*, have tubular cristae that differ in structure from that seen in mammalian cells. The mitochondrial ATP synthase complex is known to play a central role in modulating the shape of cristae. In this manuscript, Mühleip and colleagues establish the structure and structural arrangements of ATP synthase from *T. gondii*. The authors elucidate numerous divergent features of ATP synthase from *T. gondii* compared to the well-characterised ATP synthase from

mammals and yeast. These include a novel Fo subdomain that constitutes the dimerization interface, a truncated and less hydrophobic subunit-a that is part of a divergent proton translocation mechanism, and an extension of the subunit-b in the peripheral stalk that appears to take the place of the missing subunit-h of the complex. Most strikingly of all are the hexameric assemblages of the ATP synthase complex in these parasites. The authors present some evidence that these assemblages contribute to the tubular shape of mitochondrial cristae, and they demonstrate that the higher order arrangements of the complex are dependent on the presence of subunits that are restricted to apicomplexan parasites. The study provides the first high resolution structural analysis of mitochondrial ATP synthase from an apicomplexan parasite, and sets the stage for future studies to characterise the divergent features of this complex. The manuscript is well written, and the structural images and movies are quite stunning. My comments are, for the most part, minor and can be addressed by the authors with minimal additional experimentation. I am not expert in structural determination, so my comments focus on aspects of the paper that explore the role of ATP synthase in mitochondrial biology of *T. gondii*.

Major comments

1. A lot of interesting data that is pertinent to the story is buried in supplementary data. Unless Nature Communications has a limit on figure numbers (which I don't think it does), I strongly suggest moving some parts of the supplementary figures to the main body of the text. In particular, data from the ATPTG11 knockout (parts of Supplementary Figures 9 and 10 and Movie S3), which provide an experimental test of hypotheses generated from the structural data, should be included in the main body of the text. Other aspects of the supplementary data that could be moved to the main body of the text include parts of Supplementary Figure 4, which highlight the apicomplexan-specific proteins at the dimer interface of the complex.

As suggested, we have moved the panel showing parasite-specific and conserved subunits in the dimer to Figure 1b. Likewise, the tomography work comparing parental and ATPTG11-KO side-by-side is now part of main Figure 7 along with new data characterizing ATPTG11-KO growth.

2. Regarding the studies on the ATPTG11 knockout, the data clearly show that loss of ATPTG11 impairs assembly of the ATP synthase multimeric arrangements. However, I don't get a strong sense of the effects that this has on cristae architecture. The manuscript would be strengthened by data that quantify, or better describe, the changes to cristae architecture that have occurred (e.g. measuring changes in length of the cristae). Also, the authors should provide additional data on whether loss of ATPTG11 leads to any defects in parasite proliferation or mitochondrial ATP synthesis, and comment on what these data mean for the role of cristae architecture/higher order assemblages of ATP synthase for parasite proliferation.

Following the reviewer's comments, we quantified the number of cristae cross sections per mitochondrial area, which is reduced in the mutant, compared to the parental line (new suppl. Fig 13b). This shows that the internal membrane

structure is altered, whereas fluorescent microscopy and flow cytometry showed that organellar ultrastructure and membrane potential appear unaffected (suppl. Fig 13d). We performed a competitive growth assay to assess effects on parasite proliferation and found that the relative abundance of ATPTG11-KO is reduced significantly over continued passages when in mixed population with the parental line (page 12, lines 364-368; Figure 7e).. Finally, we suggest that the role of ATP synthase oligomerisation is to ensure localization of the enzyme to the tips of cristae, where high membrane potential favours ATP synthesis (page 14, lines 403-405). The two methods available for comparative quantification of three-dimensional cristae architecture would involve either large tomographic datasets and substantial segmentation work or state-of-the-art super-resolution fluorescence microscopy, both of which will constitute comprehensive studies on their own and are outside the scope of this study.

Minor comments

3. General comment on the introduction: a brief introduction on the different domains of ATP synthase (Fo, F1, rotor, stator, central stalk, peripheral stalk, etc) would help non-expert readers understand some of the later discussion. We have moved a panel to new Figure 1b, with a color scheme indicating the conserved F_o subunits in the membrane and the F₁/c-ring complex.

4. Page 2, Line 43. “to” not “do” (typo)

This typo has been corrected.

5. Page 2, Line 48. “differs substantially in its cristae morphology” from what? It reads like the comparison is to Plasmodium, whereas I think the authors mean the ‘bulbous’/tubular cristae that occur in apicomplexans compared to the flattened/lamellar cristae seen in yeast or mammalian cells (as noted in the subsequent sentence). Suggest rewording.

We have reworded the sentence to correct this.

6. Page 2, Line 49. “bulbous cristae”. Is the term bulbous used beyond the Ferguson study that is cited? The term “tubular” is more common in the literature around cristae morphology in eukaryotes. Are these “bulbous” cristae obviously different from, say, the tubular cristae in Plasmodium or ciliates to warrant a separate term?

The reference is the standard *Toxoplasma* textbook (Weiss et al, 2020, 3rd edition), which states regarding mitochondria: “They show the typical apicomplexan structure, with bulbous cristae.” The same terminology is also found in “Toxoplasmosis of animals and humans” (JP Dubey). Previous descriptions of *T. gondii* cristae shape vary due to difficulty in interpreting 2D sections. Our work shows that *T. gondii* cristae are indeed bulb-shaped, and crowned by ATP synthase pyramids. This is in stark contrast to the long, coiled tubular cristae found in ciliates (PMID: 27402755). Unlike in *T. gondii*, the tubular cristae in ciliates are shaped by helical ATP synthase dimer rows and possess one crista junction at either end of the tubular crista. Therefore, these terms for different types of cristae should be carefully distinguished, as

is now emphasized in page 14, lines 388-391

7. Page 2, Lines 50-51. “Loss of ATP synthase and the resulting defect in cristae abundance leads to death of the *T. gondii* ...” – the cited studies indicate that ATP synthase is important for these stages in the parasites, and the Huet study depicts changes in mitochondrial morphology upon ATP synthase loss. But neither study establishes that the growth defects result from the defects in cristae abundance as the text implies. The authors should reword this to be clearer about what these studies do, and do not, show.

We reformulated this section to state that loss ATP synthase was accompanied with parasite death and defects cristae abundance (page 2, lines 47-50).

8. Page 3, Line 70. I’m curious about the reasoning behind the “ATPTG1-17” nomenclature. What does the TG stand for? *Toxoplasma gondii*? The authors could specify this.

We now clarify on page 2 lines 71-575 that we follow the established nomenclature for the ATP synthase subunits that is the standard in the field.

9. Page 3, Line 70 and elsewhere: “apicomplexan-conserved” subunits. The authors could include additional details that provide evidence these subunits are conserved in apicomplexans (beyond the alignments shown in Supplementary Figure 8). Are they conserved across the apicomplexan lineage? And in other myzozoans? The answers may provide some insights into, for example, the likelihood that the newly identified Fo subcomplex is conserved across these lineages. Perhaps an addition to Table S2 could be used to clarify this?

In supplementary Fig. 10, we provide a sequence alignment of domains within the key linker proteins ATPTG5, 9 and 11 that are conserved among an array of apicomplexan parasites. Furthermore, we now added new Supplementary Figure 4, which shows the presence of ATPTG1-17 subunit homologs, showing they are mostly conserved in apicomplexa, but several subunits are also found in chromerids and perkinsozoa (page2, lines 68-71). We comment about the potential implications of this conservation in lines in page 2 lines 68-72.

10. Page 3, Lines 89-90. “This is mediated by a short N-terminal hydrophobic helix of ATPTG11, which is sequence-conserved in Apicomplexa (Fig. 2A,B).” Can the authors annotate ATPTG11 on the main part of Fig. 2A and in Fig 2B? I think this is the blue-green coloured subunit, but this is not obvious.

We have included the labels in a new Figure 2, highlighting the ATPTG11 plug on the luminal side of the c-ring and added a new panel Figure 2c to show this better.

11. Page 5, Line 20. Where the unusual dimer architecture of the *T. gondii* ATP synthase is first mentioned, authors could consider briefly describing what the conventional dimerisation interface is comprised of, so that the reader has a better appreciation for the novelty of the data.

We now clarify further that the offset peripheral stalks in *T. gondii* “does not allow the formation of the conventional dimerization interface of type-I ATP synthases found in animals and yeast (Supplementary Fig. 5i,j), in which peripheral stalks extend along the dimer long axis”, and the differences to other dimerization types are illustrated in the referenced figure.

12. Page 5, Line 37. The authors state that the inhibitor protein of ATPase activity, IF₁, contributes to the F_o dimer interface. Does this mean that dimerisation depends on IF₁, or that ATP synthase can dimerise in the absence of IF₁, but that IF₁ inhibits ATP synthase by binding along the dimer interface to lock the two F₁ regions into the ADP bound state? Or is it not possible to make any functional conclusions about the role of IF₁ from the data? It would help readers understand the importance of these data by placing this into some more context.

To clarify this point, we added a panel showing the large apicomplexan dimer interface in Figure 1b, and the minor contribution of IF₁ is shown in Figure 3b. We also calculated and stated the dimer interface that is >7,000 Å² in the context (page 5, line 126-128). In the same context we now sharpened the discussion on IF₁ to emphasize that it binds to both F₁ and F_o, thereby locking the rotor in one rotational state (page 6, lines 151-165). Finally, we deleted the mention of IF₁ from the title of Figure 3, so that it is not misinterpreted as IF₁ forming the dimer.

Together, we hope it helps to clarify the involvement of IF₁, and addresses the Reviewer’s concern.

13. Page 6, Lines 67-68. “A similar observation for different mitochondrial membrane proteins has been proposed to enable mitochondrial protein targeting following gene transfer to the nucleus (24, 25).” Also proposed for the Cox2a and 2b sequences of apicomplexan parasites – PMID 12843377.

We thank the reviewer for pointing out this reference, which we have added to the manuscript.

14. Page 8, Line 59. “contacta with” – “contacts”?

We have corrected this typo.

15. Legend, supplementary Figure 8. What is AK88 (line 75) in the ATPTG9 alignment?

This is a locus string in *P. fragile* gene names (*Pf* is already used for *P. falciparum*). We now use *Pfr* to avoid confusion.

16. Movie S2. The movies are stunning, and a strong addition to the manuscript. On my version of this movie, the ‘legend’ depicting hexamers and pentamers flashes up only very briefly – I think it would be helpful for this to appear for a longer time so they can be viewed more easily.

Thank you for the compliment and for checking these important details. The movie has been adjusted to pause longer at both label displays.

17. Page 11, Line 20. "...enabling the generation of mitochondrial mutants that are otherwise lethal". The knockout in the BCKDH-E1a subunit to which the authors refer can still proliferate in glucose-free medium (albeit more slowly than in complete medium). I don't think it accurately reflects the literature to say that mitochondrial mutants are "otherwise lethal". It would be more accurate to state that parasites "... can be cultured using alternative energy sources to OxPhos, thereby enabling the mutation of genes encoding proteins involved mitochondrial energy metabolism."

We have reformulated this passage along the suggested lines.

18. As a follow-up point, the authors don't show that ATPTG11 knockout leads to defects in ATP synthesis in the mitochondrion, so the link to OxPhos that "allowed us to generate a knockout line of ATPTG11" is not established. In fact, a genome-wide CRISPR screen suggests that ATPTG11 is dispensable for tachyzoite proliferation (PMID 27594426), which will have indicated to the authors that this gene is unlikely to be essential for parasite proliferation or ATP synthesis before they went to knock this out. I suggest rewording this section. I don't think the authors need to justify this experiment by linking ATPTG11 to OxPhos – the rationale to study the role of ATPTG11 in maintaining cristae architecture is sufficient.

We deleted the unnecessary justification, and reformulated according to the suggestion (page 12, lines 342-344).

19. Supplementary Figure 10D. The authors should indicate where the ATP synthase dimer occurs on the western blot. I believe this should be the ~1048 MDa complex, while readers might think this is the ~720 kDa complex, since a monomeric band is more visible in the KO than in the parental strain.

We have labeled the 1048-kDa band as dimer in Supplementary Fig. 12.

20. Page 33, Supplementary Figure 9, Lines 91-92. "(I-J) thin sections of *T. gondii* with mitochondria outlined (red dashed lines), showing cristae morphology." The authors should indicate in the figure legend which image is of WT mitochondria and which is of ATPTG11 mitochondria.

We now added this information as a panel label and rearranged the figure, so that the panels are shown side-by-side for readers' convenience (Figure 7, Supplementary Figure 13).

21. Page 12, Line 88. The sgRNA listed here appears to include the PAM site, which is not a component of the actual sgRNA.

We thank the reviewer for pointing this out and have removed the PAM site from the sequence.

22. Page 13, Lines 17-18. "The supernatant was added sample buffer" – "Sample buffer was added to the supernatant" perhaps?

We thank the reviewer for spotting this and have corrected this.

REVIEWERS' COMMENTS

Reviewer #1 (Remarks to the Author):

I am satisfied with the revisions and favour publication in Nature Communications.

Reviewer #2 (Remarks to the Author):

All issues brought up in the first round of review have been addressed satisfactorily.

Reviewer #3 (Remarks to the Author):

The authors have made all the changes requested.

Reviewer #4 (Remarks to the Author):

The authors have addressed all my comments and suggestions on the first draft in their revision. I have one minor follow-up question to one of their changes (on lines 403-405), and some minor grammatical/typographical suggestions. All up, this is a terrific, visually attractive paper that breaks considerable new ground in the field.

Line 403-405. In response to my suggestion of including more data on the effects of ATPTG11 KO on cristae morphology, the authors have included new data on this mutant that strengthen the manuscript. They conclude this section by stating: "we suggest that the role of ATP synthase oligomerisation is to ensure localization of the enzyme to the tips of cristae, where high membrane potential favours ATP synthesis". This is an attractive idea. However, the data on the ATPTG11 KO indicate that localisation of ATP synthase away from the tips of cristae (Figure 7b,d) results in, at best, a subtle phenotype in parasite growth (Figure 7e-f) and no defects in mitochondrial membrane potential (Supplementary Figure 13c). If the authors' hypothesis about ATP synthase multimerisation and high membrane potential contributing to ATP synthesis is true, this would imply a minimal role for mitochondrial ATP synthesis in *T. gondii* tachyzoites (or imply, contrary to their hypothesis, that the localisation of ATP synthase to tips of cristae is not particularly important for ATP synthesis). The Huet et al (2018) study found that loss of subunit b resulted in a strong defect in parasite growth and presents some data that loss of the b subunit leads to impaired mitochondrial ATP synthesis (thus implying a role for ATP synthesis in parasite growth). The authors could comment on these discrepancies and better integrate their findings into the existing literature.

Minor comments

Line 264. "...the augmented *T. gondii* subunit-b are essential ..." – a more appropriate reference here would be Huet et al (Ref 14), which demonstrated a range of defects when subunit-b was knocked down in *T. gondii* parasites.

Line 367. "results" not "result"

Line 425. "in DME medium" or "in DMEM"; "supplemented with" rather than "complemented with"

Line 426. %ages should state v/v or w/v (also applies elsewhere in methods). In the case of Pen/Strep, better to list concentrations of each antibiotic

Line 532-533. "Wet transfer ... was performed"

Line 657. "ATPTG1/cytochrome-c" – I think the authors mean "ATPTG1/cytochrome-c1"

Supplementary Figure 4. This table suggests that ATPTG5 and ATPTG11 lack homologues in many other apicomplexans, but a later supplementary figure (Supp Figure 10) shows alignments of these proteins with homologues in those other apicomplexans. Is it just that the homology scores were too low to pick up hits, whereas more sophisticated search algorithms could identify these? The authors could provide a brief clarification here.

Reviewer #4 (Remarks to the Author):

The authors have addressed all my comments and suggestions on the first draft in their revision. I have one minor follow-up question to one of their changes (on lines 403-405), and some minor grammatical/typographical suggestions. All up, this is a terrific, visually attractive paper that breaks considerable new ground in the field.

Line 403-405. In response to my suggestion of including more data on the effects of ATPTG11 KO on cristae morphology, the authors have included new data on this mutant that strengthen the manuscript. They conclude this section by stating: “we suggest that the role of ATP synthase oligomerisation is to ensure localization of the enzyme to the tips of cristae, where high membrane potential favours ATP synthesis”. This is an attractive idea. However, the data on the ATPTG11 KO indicate that localisation of ATP synthase away from the tips of cristae (Figure 7b,d) results in, at best, a subtle phenotype in parasite growth (Figure 7e-f) and no defects in mitochondrial membrane potential (Supplementary Figure 13c). If the authors’ hypothesis about ATP synthase multimerisation and high membrane potential contributing to ATP synthesis is true, this would imply a minimal role for mitochondrial ATP synthesis in *T. gondii* tachyzoites (or imply, contrary to their hypothesis, that the localisation of ATP synthase to tips of cristae is not particularly important for ATP synthesis). The Huet et al (2018) study found that loss of subunit b resulted in a strong defect in parasite growth and presents some data that loss of the b subunit leads to impaired mitochondrial ATP synthesis (thus implying a role for ATP synthesis in parasite growth). The authors could comment on these discrepancies and better integrate their findings into the existing literature.

We’ve added a discussion on this point on page 8, lines 310-316 and integrated the findings with the previous work Huet et al 2008. The added paragraph describes that ATPTG11-KO selectively disrupts higher oligomer formation, which interferes with its membrane-shaping function and macromolecular localization, and thus presumably leaves the assembled dimer units catalytically competent. This is different from the loss of the core subunit *b* described in Huet et al 2008 that has been shown to interfere with ATP synthase assembly, implying abolishment of its catalytic activity. Therefore, the mild phenotype detected in our study may be a result of a larger ATP synthase fraction locating to the inner boundary membrane, where it cannot harvest the electrochemical potential efficiently.

Minor comments

Line 264. “...the augmented *T. gondii* subunit-b are essential ...” – a more appropriate reference here would be Huet et al (Ref 14), which demonstrated

a range of defects when subunit-b was knocked down in *T. gondii* parasites.
We have added this reference.

Line 367. “results” not “result”

Fixed

Line 425. “in DME medium” or “in DMEM”; “supplemented with” rather than “complemented with”

Fixed.

Line 426. %ages should state v/v or w/v (also applies elsewhere in methods).
In the case of Pen/Strep, better to list concentrations of each antibiotic

We corrected this.

Line 532-533. “Wet transfer ... was performed”

Fixed.

Line 657. “ATPTG1/cytochrome-c” – I think the authors mean “ATPTG1/cytochrome-c1”

Fixed

Supplementary Figure 4. This table suggests that ATPTG5 and ATPTG11 lack homologues in many other apicomplexans, but a later supplementary figure (Supp Figure 10) shows alignments of these proteins with homologues in those other apicomplexans. Is it just that the homology scores were too low to pick up hits, whereas more sophisticated search algorithms could identify these? The authors could provide a brief clarification here.

As suggested, we added a clarification in the legend of the Suppl Fig. 4 that it represents an overview of subunit conversion within different apicomplexan subgroups, for some of which homologs could not be identified based on tBLASTn searches against the respective *T. gondii* gene.

As the reviewer suggests, the sequence conservation is highest within the closely related family of *Sarcocystidae*, for which several homologs are shown in Suppl Fig. 10e,f (*Sn*, *Nc*, *Hh*, *Cs*). It shows that several *Plasmodium* homolog candidates align well to this set of conserved sequences, showing local conservation (e.g the N-terminal helix of ATPTG11), including the critical cysteine pairs identified in the structure (Suppl. Fig. 10). We therefore included these low-scoring hits as *bona-fide* candidates for homologs.

Indeed, as the reviewer noticed, *H. hammondii* was included as a representative of *Sarcocystidae* homologs in Suppl. Fig. 4 to avoid bias

towards this subgroup, even though more homologs are readily identified.
This is now clarified in the figure legend.